# Quantification reveals early dynamics in *Drosophila* maternal gradients

**Alex Shlemov[1], Theodore Alexandrov[2,3], Nina Golyandina[4], David Holloway[5], Stefan Baumgartner[6,7], Alexander V. Spirov[8,9]***

**1** Laboratory for Algorithmic Biology, St. Petersburg State University, St. Petersburg, Russia, **2** Structural and Computational Biology Unit, European Molecular Biology Laboratory, Heidelberg, Germany, **3** Skaggs School of Pharmacy and Pharmaceutical Sciences, University of California San Diego, La Jolla, California, United States of America, **4** Faculty of Mathematics and Mechanics, St. Petersburg State University, St. Petersburg, Russia, **5** Mathematics Department, British Columbia Institute of Technology, Burnaby, British Columbia, Canada, **6** Department of Experimental Medical Sciences, Lund University, Lund, Sweden, **7** Department of Biology, University of Konstanz, Konstanz, Germany, **8** Computer Science and CEWIT, SUNY Stony Brook, Stony Brook, New York, United States of America, **9** Lab Modelling Evolution, The I.M. Sechenov Institute of Evolutionary Physiology & Biochemistry, St. Petersburg, Russia

* alexander.spirov@gmail.com

**Data Availability Statement:** The data underlying the results presented in the study are within the paper and its Supporting information files. Additionally, the data and scripts for the figures and

## Abstract

The Bicoid (Bcd) protein is a primary determinant of early anterior-posterior (AP) axis specification in *Drosophila* embryogenesis. This morphogen is spatially distributed in an anterior-high gradient, and affects particular AP cell fates in a concentration-dependent manner. The early distribution and dynamics of the *bicoid* (*bcd*) mRNA, the source for the Bcd protein gradient, is not well understood, leaving a number of open questions for how Bcd positional information develops and is regulated. Confocal microscope images of whole early embryos, stained for *bcd* mRNA or the Staufen (Stau) protein involved in its transport, were processed to extract quantitative AP intensity profiles at two depths (apical—under the embryo surface but above the nuclear layer; and basal–below the nuclei). Each profile was quantified by a two- (or three-) exponential equation. The parameters of these equations were used to analyze the early developmental dynamics of *bcd*. Analysis of 1D profiles was compared with 2D intensity surfaces from the same images. This approach reveals strong early changes in *bcd* and Stau, which appear to be coordinated. We can unambiguously discriminate three stages in early development using the exponential parameters: pre-blastoderm (1–9 cleavage cycle, cc), syncytial blastoderm (10–13 cc) and cellularization (from 14A cc). Key features which differ in this period are how fast the first exponential (anterior component) of the apical profile drops with distance and whether it is higher or lower than the basal first exponential. We can further discriminate early and late embryos within the pre-blastoderm stage, depending on how quickly the anterior exponential drops. This relates to the posterior-wards spread of *bcd* in the first hour of development. Both *bcd* and Stau show several redistributions in the head cytoplasm, quite probably related to nuclear activity: first shifting inwards towards the core plasm, forming either protrusions (early pre-blastoderm) or round aggregations (early nuclear cleavage cycles, cc, 13 and 14), then moving to the embryo surface and spreading posteriorly. These movements are seen both with the 2D surface study and the 1D profile analysis. The continued spreading of *bcd* can be tracked

tables are contained in the Zenodo repository at https://doi.org/10.5281/zenodo.5155678 (Shlemov Alex, Alexandrov Theodore, & Golyandina Nina. (2021). "Quantification reveals early dynamics in Drosophila maternal gradients" (v.0.1)).

**Funding:** This work was supported by A.V.S.: the Russian Foundation for Basic Research grant 20-04-01015 (https://www.rfbr.ru/); N.G. and A.S.: the Russian Foundation for Basic Research grant 20-01-00067; S.B.: the Swedish Research Council 2010 (https://www.vr.se/), and the Ekhaga, Nilsson-Ehle and Erik Philip-Sörensen Foundation. The funders had no role in study design, data collection and analysis, decision to publish, or preparation of the manuscript.

**Competing interests:** The authors have declared that no competing interests exist.

from the time of nuclear layer formation (later pre-blastoderm) to the later syncytial blastoderm stages by the progressive loss of steepness of the apical anterior exponential (for both *bcd* and Stau). Finally, at the beginning of cc14 (cellularization stage) we see a distinctive flip from the basal anterior gradient being higher to the apical gradient being higher (for both *bcd* and Stau). Quantitative analysis reveals substantial (and correlated) *bcd* and Stau redistributions during early development, supporting that the distribution and dynamics of *bcd* mRNA are key factors in the formation and maintenance of the Bcd protein morphogenetic gradient. This analysis reveals the complex and dynamic nature of *bcd* redistribution, particularly in the head cytoplasm. These resemble observations in oogenesis; their role and significance have yet to be clarified. The observed co-localization during redistribution of *bcd* and Stau may indicate the involvement of active transport.

## 1 Introduction

The concept of morphogenetic gradients in embryo development was introduced in the form of "physiological" gradients of activity many decades ago [1]. During the 1960s and early 1970s, this concept was refined in terms of substances (morphogens) which diffused across fields of cells to form spatial concentration gradients [2–4]. Simple molecular mechanisms such as localized protein synthesis, diffusion and degradation can produce stationary non-uniform distributions of morphogens that govern embryonic pattern formation. A number of morphogenetic gradients have been studied in detail experimentally; the most heavily studied of these is the Bicoid (Bcd) protein gradient along the anterior-posterior (AP) axis of the fruit fly (*Drosophila*) embryo [5, 6].

The Bcd protein gradient arises from maternally deposited *bcd* mRNA at the anterior pole of the very early embryo [7]. *bcd* mRNA is translated at the start of egg deposition, and the protein gradient is established within 3 hours. The Bcd morphogenetic gradient is subsequently read by ~20–30 target genes, which are expressed in discrete patterns along the AP axis [8–13].

Complex molecular machinery provides transport and localization of mRNA in the oocyte [14–20]. *bcd* mRNA is deposited compactly at the tip of the zygote's head in a conglomerate, complexed with several proteins [21–25]. This forms a highly specialized structure, supporting sustained translation and the production of large amounts of Bcd protein (Cf [26, 27]). While many details of how this structure operates are still unclear (see [28]), a widely-used working hypothesis has been to treat the mRNA as a point source (with negligible spatial extent) for Bcd protein [29, 30]. This leads to a Synthesis-Degradation-Diffusion (SDD) model for the exponential-in-distance concentration gradient of Bcd, in which synthesis occurs at the anterior point, diffuses posteriorly and decays [31, 32].

Other morphogens have been discovered and characterized since Bcd (e.g. [33, 34]). These display a number of complicated and redundant regulatory networks, likely contributing to robustness to external perturbances and intrinsic fluctuations, which make the SDD model look somewhat suspiciously simple (Cf [28]). More recent investigations into Bcd hint at a deeper complexity here as well (Cf [35]). In particular, evidence is emerging that *bcd* mRNA forms an extended gradient which is highly dynamic in space and time, with strong implications for the patterning mechanism of the morphogenetic Bcd protein gradient [36–40].

While *bcd* RNA is tightly bound to the anterior cortex of the developing egg, it has been known since the late 1980s that its localization becomes more diffuse after fertilization [7]. Results from [18] suggest that release of *bcd* from the cortex involves an activation-dependent

restructuring of the actin cytoskeleton. Egg activation also triggers *bcd* mRNA redistribution outside the processing bodies (P-bodies), regions of the cytoplasm where regulation of translation and mRNA degradation occur [26]. Weil et al. [26] proposed that translational activation of *bcd* mRNA results from the disassembly of P-bodies on egg activation.

Several early publications (reviewed by [41]) stated that *bcd* mRNA forms an AP concentration gradient prior to the cellular blastoderm stage [42]. Studies on the Bcd protein gradient overshadowed these results, but problems with the SDD model (e.g. scaling to variable embryo length [43]), revived interest in mRNA patterning. Using a more sensitive fluorescence *in situ* hybridization (FISH) technique, Spirov et al. [36] demonstrated that (1) a *bcd* mRNA concentration gradient is formed along the cortex of the embryo by the beginning of the syncytial blastoderm stage (cc10); (2) the gradient falls off exponentially with distance from the anterior pole and persists unchanged during cc10-13; (3) *bcd* is co-localized with the Staufen (Stau) factor, which indicates the active nature of the *bcd* redistribution; (4) *bcd* mRNA is transported to the apical nuclear periplasm during syncytial blastoderm stages; and (5) *bcd* transcripts are degraded rapidly during the first third of cc14.

Little et al. [44] used a novel mRNA quantification method, visualizing individual mRNA particles in whole embryos, to further investigate the dependence of Bcd protein gradient formation on the strength, localization, and dynamics of the underlying *bcd* mRNA. They determined that >90% of all *bcd* mRNA is continuously present within the anterior 20% of the embryo (i.e. all but a few mRNA particles are confined to the anterior 20% of the egg). They concluded that although the spatial extension of the mRNA contributes to shaping the protein gradient, mRNA localization alone cannot account for protein gradient formation. These results were used to formulate an extended SDD model [44].

Cheung et al. [45] showed that *bcd* mRNA plays a role in length scaling, with large embryos having more maternally deposited *bcd* mRNA than small embryos. They [46] also found anomalous cases of larger embryos with atypically broad *bcd* distributions. With sequencing, they found that these embryos had mutations in the coding regions of *Vps36* and *stau*, each of which encode proteins associated with *bcd* mRNA transport and redistribution during oogenesis and early embryo development, and indicating that these processes are part of *bcd* patterning and scaling.

More recently, Ali-Murthy and Kornberg [38] have corroborated the Spirov et al. [36] and Little et al. [44] observations on the *bcd* mRNA distribution in the syncytial blastoderm stage embryos. The authors paid special attention to *bcd* redistribution during the pre-syncitial stages (nc1 –nc6). They observed that *bcd* mRNA and Bcd protein formed similar distributions in the interior of pre-syncytial blastoderm embryos.

*bcd* mRNA molecules are very heavy (heavier than Bcd protein molecules), persisting through oogenesis and early embryogenesis in the form of large ribo-nuclear protein (RNP) complexes. Such macromolecular aggregates cannot diffuse effectively and likely require special mechanisms for transport. The ARTS (active RNA transport and protein synthesis) model (in contrast to SDD) proposes that such a transport mechanism, e.g. migration along microtubules (MTs), plays a role in formation of the *bcd* mRNA gradient in oogenesis and early embryogenesis [36].

Using a modified permeabilization and fixation method to rapidly fix and preserve cortical MT structures, Fahmy et al. [37] indeed detected MT activity in early cleavage cycle embryos and demonstrated that there is an extensive MT network exclusively in the anterior half of the cortex. Further, they show that αTubulin67C and the kinesin-like minus-end motor Ncd are critically important for transport of *bcd* mRNA in the oocyte and along the embryonic cortex to establish the mRNA gradient.

Ali-Murthy and Kornberg [38] concluded that the complex and well-defined shapes and the rapidity with which the distributions form and change would seem to be incompatible with passive diffusion. They hypothesized that motor-driven directed movement might transport particles of *bcd* RNA (Cf [36, 37]) and protein posteriorly to form the internal plume in pre-cc6 embryos, and to re-distribute bcd RNA and protein to the cortex at blastoderm stages.

Recently, Cai et al. [39] demonstrated the existence of at least two distinct and undiscovered *bcd* transport systems in the early embryo: a cortical path and a core path. The cortical path relies critically on aTub67C. The core path involves an internal "core plasm" system which rapidly transports *bcd* mRNA to the interior of the embryos; this path does not require functional aTub67C nor Ncd. The cortical path is built up only after fertilization [37]; the core mechanism can transport *bcd* mRNA in unfertilized eggs.

The new evidence on a role for *bcd* mRNA in Bcd gradient formation has spurred a number of modelling studies regarding the gradient mechanism, i.e. whether gradient formation entirely follows from mRNA dynamics (e.g. the ARTS model: [36, 37, 39, 40, 47]), or if there is a substantial role for Bcd protein diffusion and decay in gradient formation [44, 48, 49].

While a number of modelling projects address details and processes not yet quantified experimentally [44, 48, 50–59]), many are indicating that the extended mRNA source is important, and that the SDD model is too simple.

These modelling studies begin to show the range of potential mechanisms for *bcd* mRNA dynamics and their effect on the Bcd protein morphogenetic gradient. They show the need for new quantitative data and data analysis to test the models to achieve a deeper understanding of the gradient forming dynamics.

There are still critical gaps regarding the shape of the *bcd* mRNA spatial distribution and its temporal dynamics in the first few hours of development, and these gaps impede our understanding of the establishment of this primary morphogenetic gradient (Fig 1). While it is known that *bcd* mRNA forms an AP concentration gradient, researchers do not yet agree on how sharp this gradient is (with a shallower gradient corresponding to a more broadly distributed Bcd production area). The shallower the *bcd* mRNA gradient, the lower the Bcd diffusivity required to produce the observed Bcd morphogenetic gradient (Fig 1A and 1B). Extension of the *bcd* mRNA domain posteriorly with each nuclear cycle could ensure the necessary growth in Bcd production during development (Fig 1C and 1D). Finally, for scalability, larger embryos need more Bcd protein, and hence more *bcd* mRNA (Fig 1E and 1F).

In addition, *bcd* RNA continuously redistributes in the volume of the maturing oocyte and developing zygote. The role of some of these redistributions remains unclear, but appear to involve active transport mechanisms and pre-cellular structures in the cytoplasm. New and larger datasets are needed to achieve sound quantitative conclusions. A complete study will need not only the new data, but new computational tools for processing, analysis and modelling the data. In this publication we present new approaches for mathematical modelling and data analysis for the expression profiles (1D) and surfaces (2D) of *bcd* RNA, and compare these results with expression of Stau.

A big challenge for collecting such data is the highly dynamic, variable and noisy patterning of *bcd* mRNA and associated early factors, such as Stau, see S1 Fig (and [36]). We recently described new methods for extracting and quantifying *bcd* and Stau data to address such variability [60]. Using a larger dataset than in previous publications, we extract data for both 1D AP intensity profiles and 2D intensity surfaces. The 2D data can help to confirm results from data analysis in 1D, in which AP profiles are quantified as two- (or three-) exponential functions (see Fig 2). We find that:

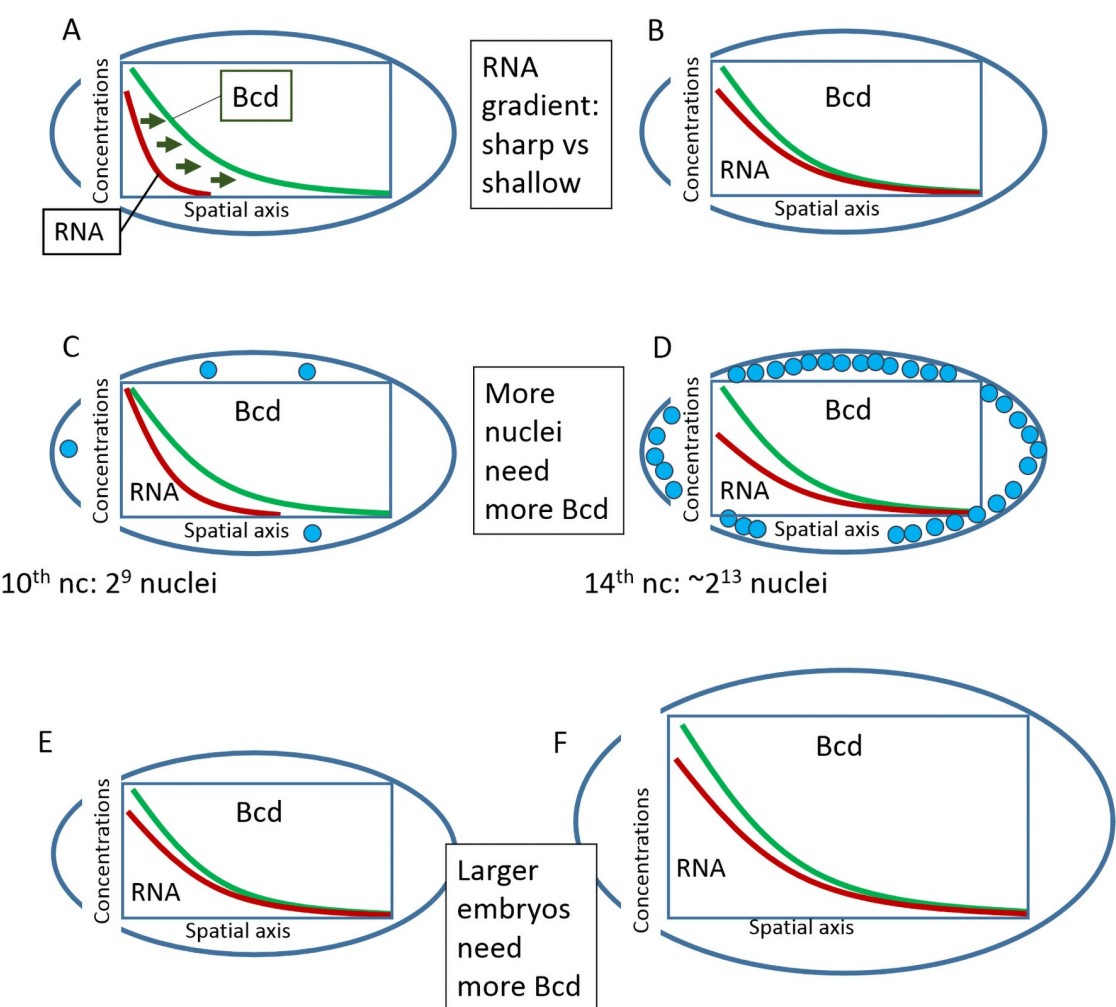

**Fig 1. Some open problems regarding the production area of Bcd protein in the early embryo.** A-B. The more extended the Bcd production area, the lower the Bcd diffusivity needs to be. Short sources require fast diffusion for the Bcd molecules to form the morphogenetic gradient (A), while an extended source makes the gradient less dependent on fast Bcd diffusion (B). C-D. An adaptable Bcd production area could compensate for required changes in Bcd production. Bcd is mainly sequestered by nuclei, so nucleus doubling in each cycle could require more total Bcd, or higher Bcd production. E-F. Large embryos should have more extended and powerful Bcd production areas.

1. Three early stages (pre-blastoderm, 1–9 cc; syncytial blastoderm, 10–13 cc; and cellularization, 14 cc) can be unambiguously discriminated by the exponential parameters, depending on how sharply the anterior exponential drops and whether this component is higher in apical or basal layers of the embryo. The results are similar for *bcd* mRNA and Stau protein.

2. The sharpness of the anterior decline can be used to distinguish early and late embryos in the pre-blastoderm stage, which relates to the spreading of *bcd* mRNA posteriorly in the first hour of development (though this estimate of mRNA extension is sensitive to background in the signal).

3. Both *bcd* mRNA and Stau show several redistributions in the head cytoplasm, probably related to nuclear activity and / or cytoplasmic fountain streams: first shifting inwards

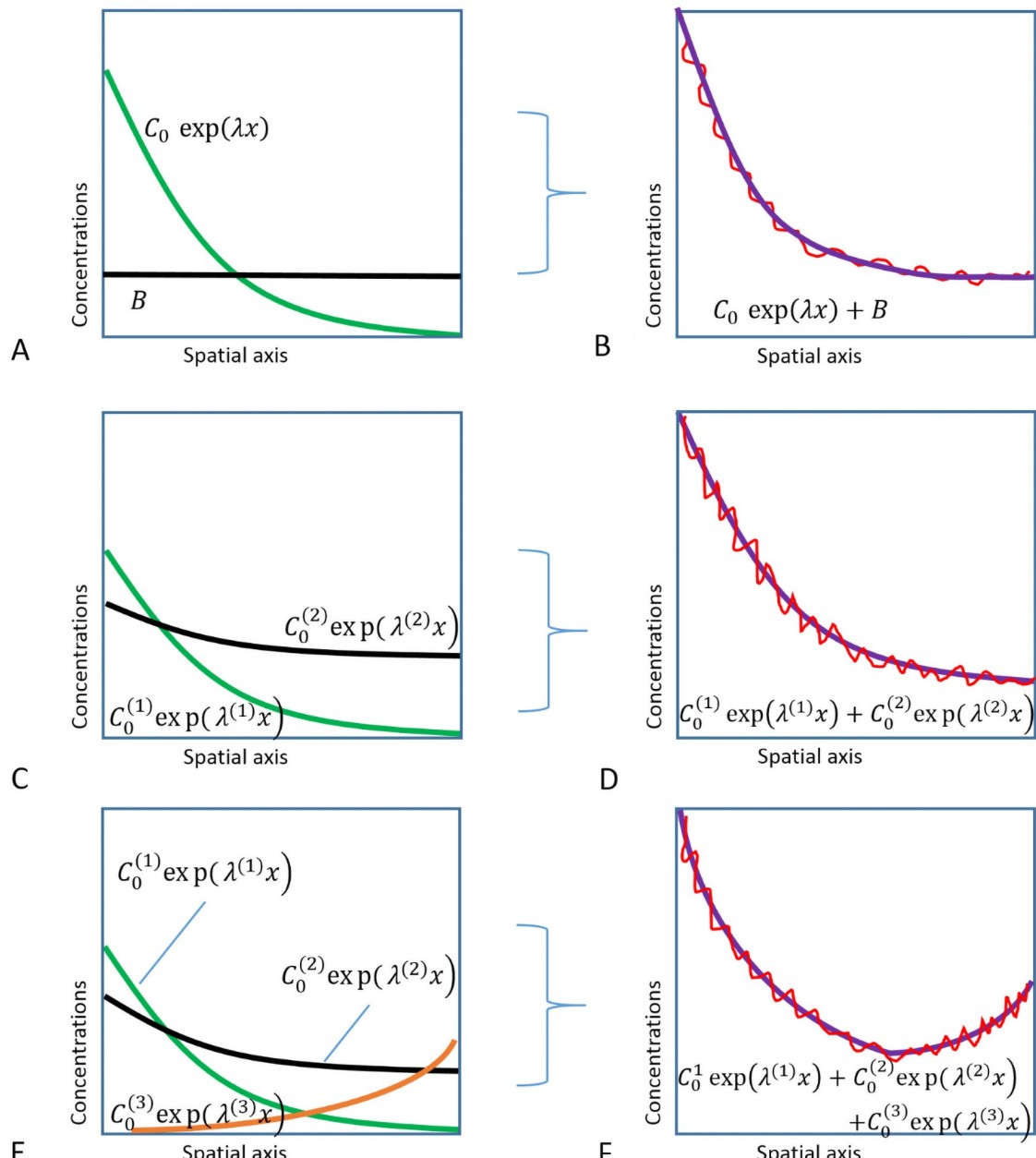

**Fig 2. Approximating (purple, right) 1D anteroposterior experimental data profiles (red, right) by: (A-B) exponential (green, left) plus background (black, left); (C-D) two- (green, black); (E-F) or three-exponential (green, black, orange) models.**

towards the core plasm, forming either protrusions (early pre-blastoderm) or round aggregations (early cc 13 and 14), then moving to the embryo surface and spreading posteriorly.

4. *bcd* mRNA continues to spread from the time of nuclear layer formation (later pre-blastoderm) to the later syncytial blastoderm stages; this can be tracked by the progressive loss of steepness of the apical anterior exponential (for both *bcd* and Stau).

5. At the beginning of cc14 (cellularization stage), we see a distinctive flip from the basal anterior gradient being higher to the apical gradient being higher (for both *bcd* and Stau).

## 2 Results

For each embryo, apical and basal fluorescence intensity profiles (for *bcd* mRNA or Stau protein) were extracted via chains of circular regions of interest (ROI) from a sagittal section of a 3D microscope scan (Fig 17A; see Methods for details). As discussed in our recent technique paper [60], a two-exponential function ($C_0^{(1)}\exp\ (\alpha^{(1)}x) + C_0^{(2)}\exp\ (\alpha^{(2)}x)$ [61]; Fig 2C and 2D) provides a better fit to *bcd* mRNA profiles than the single-exponential plus uniform background ($C_0\exp(\alpha x) + B$ [30], Fig 2A and 2B) more traditionally applied to the Bcd protein gradient, and is especially useful for following the variable *bcd* mRNA profiles encountered in early developmental stages. We have also found that a three-exponential function ($C_0^{(1)}\exp\ (\alpha^{(1)}x) + C_0^{(2)}\exp\ (\alpha^{(2)}x) + C_0^{(3)}\exp\ (\alpha^{(3)}x)$) extends our ability to fit profiles with a posteriorly rising component, as is found in some cases for Stau (Figs 2E and 2F and 16). This approach allows us to quantify data profiles with reliable parameters: the exponential decay constants $\alpha$ (or the exponential base $\lambda = \exp(\alpha)$) and pre-exponential factors $C_0$. These are used to track developmental changes in the *bcd* and Stau distributions, which reflect the underlying mechanisms for transport and localization of the factors.

### 2.1 Quantitative tracking of the developmental sequence of bcd mRNA and Stau gradient changes: Three main stages

**2.1.1 Linear discriminant analysis.** With the basal and apical profiles of each embryo quantified in terms of $\lambda$s and $C_0$s, we used Linear Discriminant Analysis [62, 63] to classify the profiles. Four parameter combinations (see Methods), $\lambda_{\text{anterior}}^{(\text{apical})}$, $\lambda_{\text{anterior}}^{(\text{basal})}$, $C^{\text{ab}} = \ln\left(C_{\text{anterior}}^{(\text{apical})}/C_{\text{anterior}}^{(\text{basal})}\right)$, $C_{(\text{shalow})}^{(\text{apical})}\alpha_{(\text{shalow})}^{(\text{apical})}/\left(C_{(\text{shalow})}^{(\text{basal})}\alpha_{(\text{shalow})}^{(\text{basal})}\right)$ gave a clear separation of embryos into three groups, corresponding to the developmental stages of pre-blastoderm (cleavage, cc 1–9), syncytial blastoderm (cc 10–13) and cellularizing blastoderm (cc 14) (Fig 3). (In total we had 38 bcd vs 15 Stau cleavage stage embryos, 43 bcd versus 22 Stau syncytial stage embryos, and 56 bcd versus 18 Stau cellulization stage embryos). As shown in S1 Table in S1 File, the developmental stage is predicted correctly for a majority of *bcd* mRNA profiles: 68% for cleavage; 78% for cc10-13; and 80% for cc14 (all significant at $p < 0.01$). For Stau, correct prediction is 50%, 73%, and 69% respectively, for these stages (S2 Table in S1 File and Fig 3B; all significant at $p < 0.01$).

**2.1.2 Parameters for age classification.** The main profile features predicting the differences between developmental stages are: the exponential decay term of the major, anterior compenent $\lambda_{\text{anterior}}^{(\text{apical})}$, and $C^{\text{ab}}$, the ratio of heights (pre-exponential factors) between apical and basal profiles for this component. The first exponential component captures a large majority of *bcd* and Stau intensity data, especially at pre-blastoderm and syncytial blastoderm stages. Fig 4 shows the mean values of these parameters by developmental stage. Fig 4A (*bcd* mRNA) and 4B (Stau protein) both show sharp, localized gradients in the pre-blastoderm stage (small $\lambda_{\text{anterior}}^{(\text{apical})}$) compared to more spread out gradients (larger $\lambda_{\text{anterior}}^{(\text{apical})}$, more close to 1) at later stages. The ratio of apical to basal profile anterior intensities, $C^{\text{ab}} = \ln\left(C_{\text{anterior}}^{(\text{apical})}/C_{\text{anterior}}^{(\text{basal})}\right)$, also shows distinct trends. (This ratio is robust to a linear transformation of data; i.e. robust to the arbitrary intensity scale which can depend on experimental procedures and microscope settings; see Methods for details.) There is a relative decrease in the apical intensity going from pre-blastoderm to the syncytial blastoderm stage. For *bcd* mRNA (Fig 4A), there is a flip from apical being higher than basal, and to basal higher than apical. Similarly, Stau has a decrease in the apical intensity, but basal remains higher than apical throughout these stages (Fig 4B). Going from syncytial blastoderm to the cellularizing blastoderm stage, both *bcd* mRNA and Stau

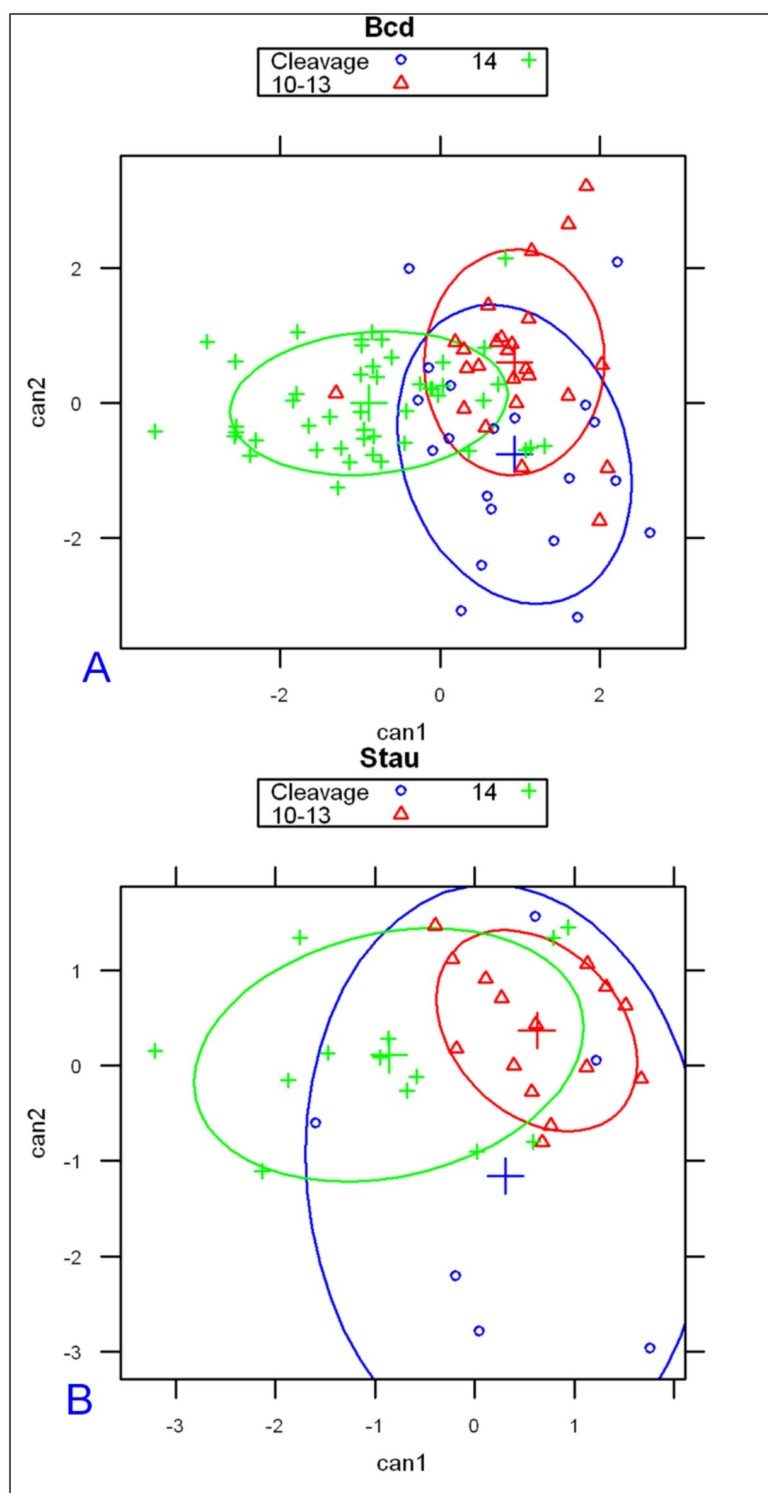

**Fig 3. Linear discriminant analysis on model parameters from *bcd* mRNA (A) and Stau (B) profiles correctly predicts developmental stage for the majority of embryos.**

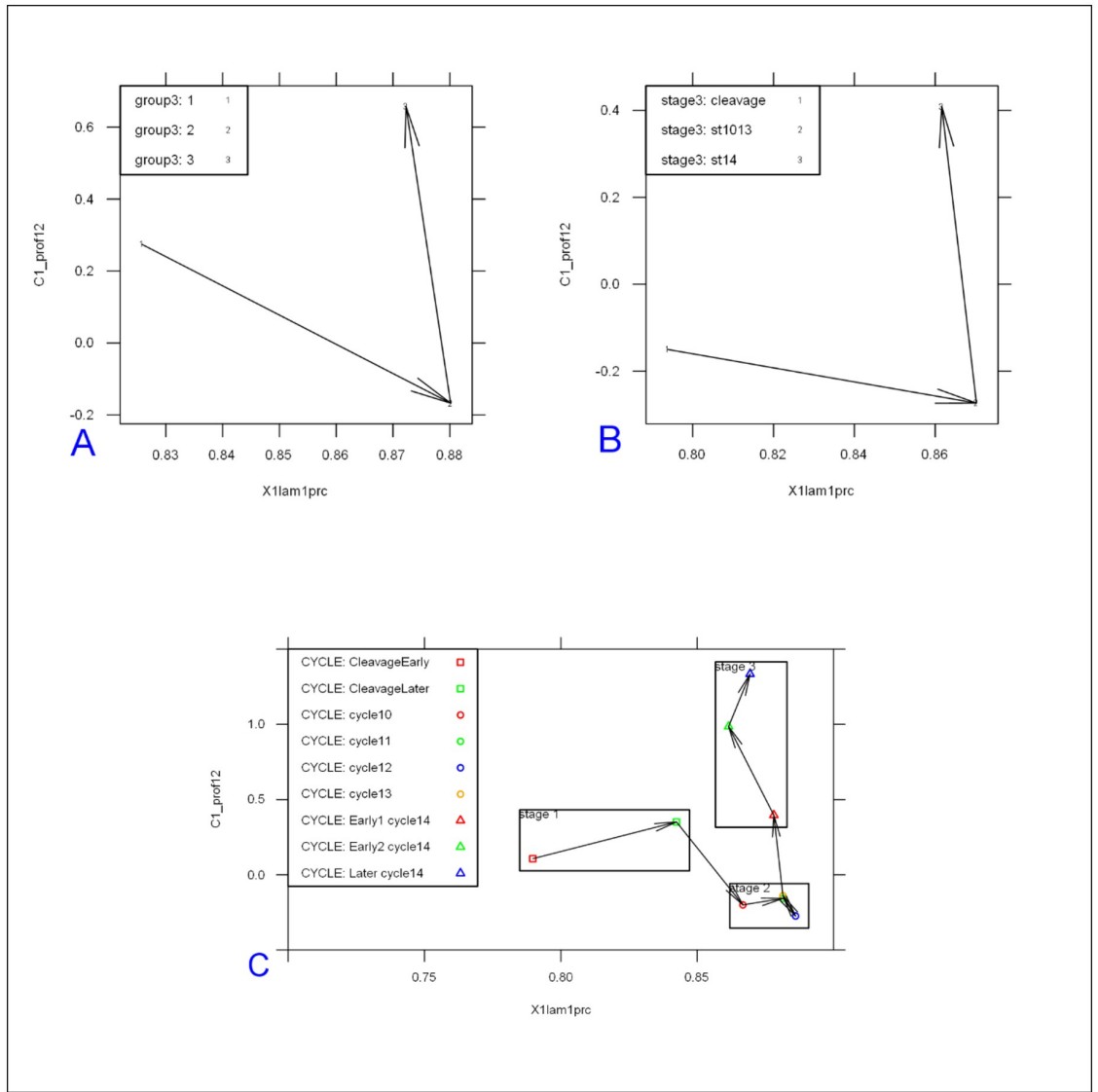

**Fig 4. The change in two-exponential model parameters in time is similar for *bcd* mRNA (A) and Stau (B): Mean values of** $\lambda_{\text{anterior}}^{(\text{apical})}$ **vs.** $C^{\text{ab}} = \ln\left(C_{\text{anterior}}^{(\text{apical})}/C_{\text{anterior}}^{(\text{basal})}\right)$**, showing distinct developmental trajectories.** Means are significantly different for the 3 main groups shown. (C) *bcd* mRNA data over nine age sub-groups.

show a strong flip, from basal being higher than apical, to apical higher than basal. The profile parameters quantify more general observations of posterior-wards spreading in earlier stages followed by a fast basal to apical movement of *bcd* mRNA and Stau in cc14, earlier described by [36].

For *bcd* mRNA, a larger dataset allows us to consider nine age sub-groups (CleavageEarly, CleavageLate, cc10, cc11, cc12, cc13, cc14early1, cc14early2, cc14late) used in [36] (see also Methods). Fig 4C indicates that the changes in profile parameters at this finer timescale follow a smooth trajectory through development.

**2.1.3 Prototypical profiles by developmental stage.** We construct prototypical profiles using the mean 2-exponential function parameters for each main stage. Results are shown in

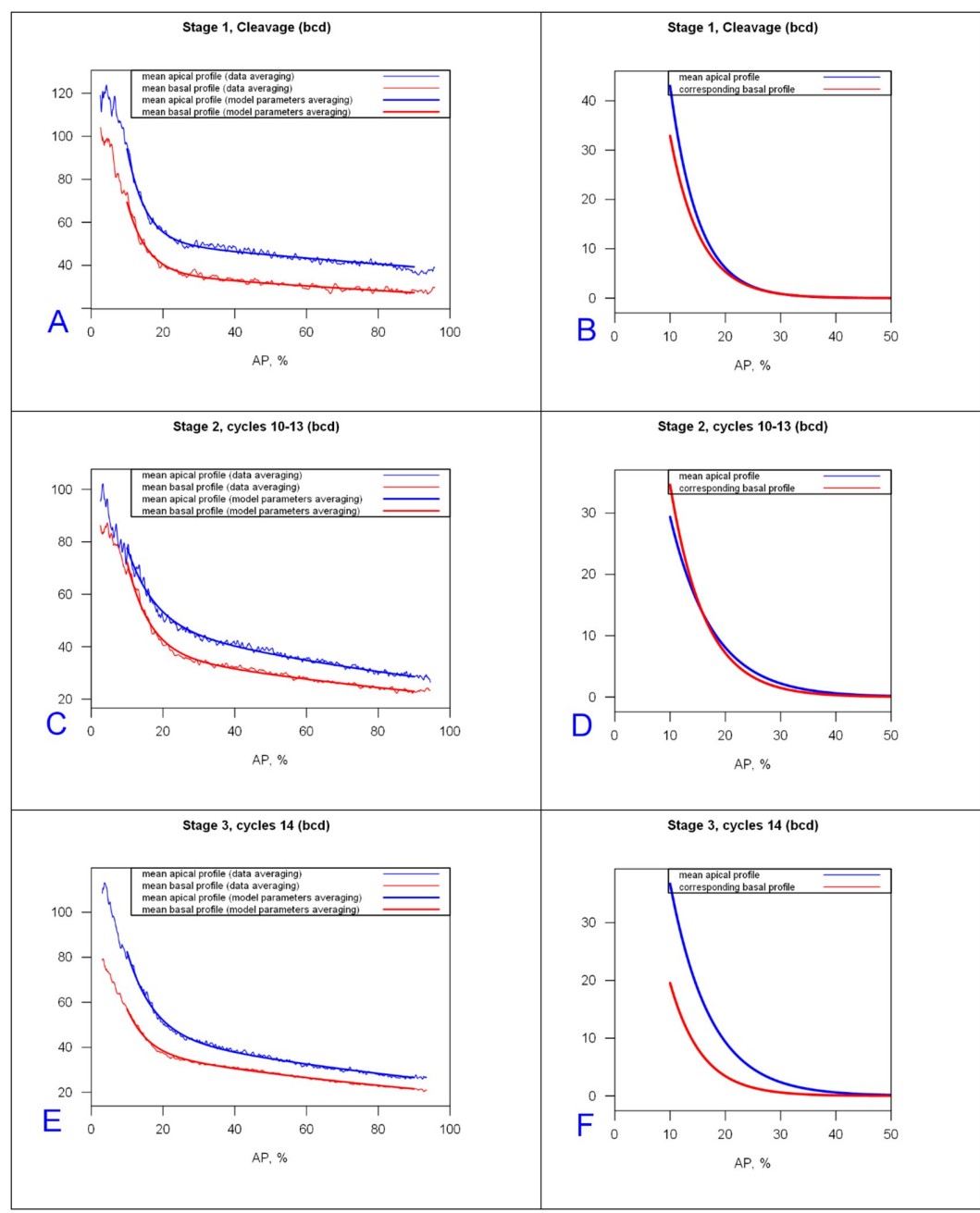

**Fig 5. *bcd* mRNA prototypical profiles and prototypical anterior exponentials for cleavage, cc10-13, and cc14 stages (fit to (10,90) %EL).** (A,C,E) profiles from mean parameters of the 2-exponential model overlaid on averaged data. Blue, apical; red, basal. (B,D,F) Anterior gradients only for the mean model profiles from (A,C,E). The anterior gradient approaches zero around 25–30%EL in B; around 30–35%EL in D and the apical profile of F; but around 20–25%EL for the basal profile in F.

Fig 5 for *bcd* mRNA (mean values in S3 Table in S1 File) and Fig 6 for Stau (S4 Table in S1 File for mean parameters).

Figs 5D–5F and 6D–6F illustrate the trends shown in Fig 4A and 4B: the increase of the exponential decay term of the major anterior component with developmental stage (broadening); and flipping of the apical and basal intensities by stage. Stau profiles are shorter than *bcd*

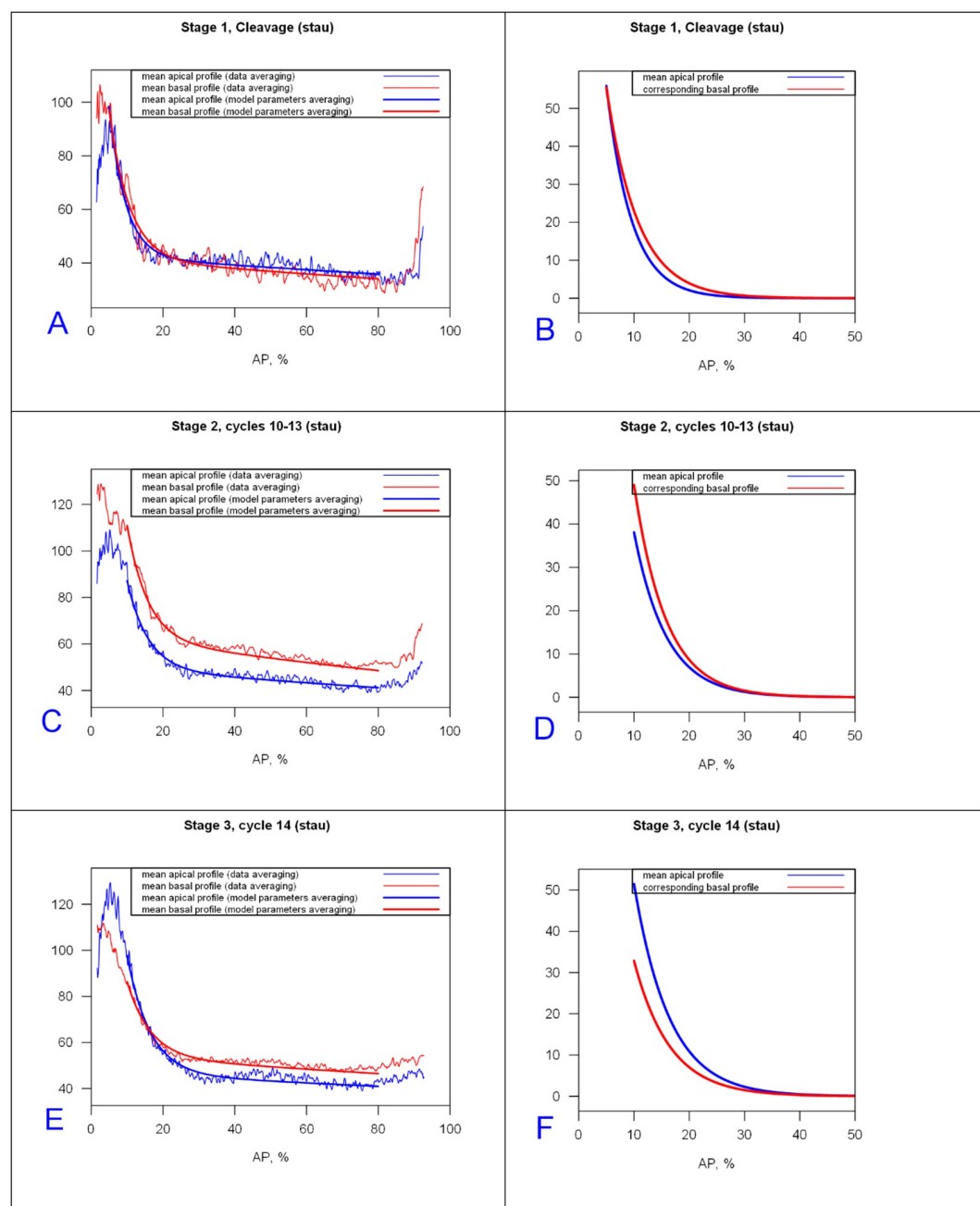

**Fig 6. Stau prototypical profiles and prototypical anterior exponentials for cleavage, cc10-13 and cc14 stages (fit to (10,80) %EL).** (A,C,E) Profiles of mean model parameters overlaid on averaged data. B,D,F) Anterior gradients only for the mean profiles in (A,C,E).

mRNA, especially in the preblastoderm stage (*Stau* has a smaller $\lambda_{\text{anterior}}^{(\text{apical})}$), and both drop off much more quickly than Bcd protein profiles [61].)

The present 2-exponential analysis shows that this occurs mainly in the second, shallow, exponential ($\lambda^{(2)}$) along with decreasing initial values of the anterior exponential ($C_0$), while the anterior exponential ($\lambda^{(1)}$) is relatively fixed and closer to 1 (Fig 5D–5F, cf S3 Table in S1 File).

Some Stau profiles show a short sharp rise in the posterior which is well-fit by a third exponential component. Being in the posterior, this component is likely not associated with *bcd*, but is more likely from Stau molecules involved in the transport of another mRNA species, such as *oskar* [64]. Stau profiles are also flatter than *bcd* in the mid- and tail parts of the embryo (Fig 5A–5C vs. Fig 6A–6C). This could indicate that either there is no Stau gradient in mid and posterior regions, or that a shallow anterior-to-posterior *bcd*-associated gradient does exist but is masked by a posterior to-anterior Stau gradient involved in active transport of a different, posterior mRNA.

Classification by exponential parameters shows similar trends for bcd mRNA and Stau protein, showing the degree to which these molecules colocalize during early embryogenesis (up to the beginning of cycle 14A).

## 2.2 Transitions within and between the main stages, from cleavage to cellularization

Here, we use the 2-exponential parameter changes to characterize in finer detail how the *bcd* mRNA and Stau profiles change during early development, within and between the main stages identified in section 2.1. The analysis of 1D profiles is then confirmed and clarified on 2D sagittal expression surfaces.

**2.2.1 Pre-blastoderm.** The first stage, Cleavage, lasts about 80 min (at room temperature). This stage is relatively long and dynamic, with *bcd* and Stau redistribution along both the AP and BA (basal-apical) axes of the sagittal plane. The early posterior spreading of *bcd* mRNA can be clearly identified within the Cleavage stage data, as seen with the increasing mean lambda1 from ClErl (up to cc7, 4–32 nuclei) to ClLat (cc 8–9, formation of the nuclear layer) in Fig 4C.

Looking at 2-exponential parameters from all the Cleavage stage data, we see that separation into Early and Late sub-stages can also stem from differences in intensity (C1_prof12; Fig 7A), with apical becoming stronger and more similar to basal in developmental time.

These posterior movements indicated by the 1D profile parameters can be seen also in Fig 8, for profiles extracted through the anterior pole, showing this spread from the initial source (note the greater extent dorsally than ventrally). The spreading corresponds to the increasing mean lambda1 from ClErl to ClLat (Figs 4C and 7).

*2.2.1.1 Redistribution in sagittal plane.* The evolution of the 1D expression profile parameters observed at the pre-blastoderm stage (the apical exponent becoming higher than the basal one) can correspondingly be seen in the redistribution of bcd and Stau in the sagittal plane (2D data). Observations and quantitative analysis demonstrate the movement of the bcd (and Stau) containing material from the head cortex into the interior of the head cytoplasm during the early cleavage divisions (we observed these processes mainly during the 4–32 nuclei stages; Cf S2 and S3 Figs). Somewhat later (later cleavage; 128–512 nuclei stages), this material turns back to the head cortical layers and spreads more broadly towards the posterior (Fig 8).

Fig 9 illustrates how the movement of the material from the head apical cortex to deep into the interior of the head cytoplasm and back to the apical cortex corresponds to changes in the relative height of the 1D apical and basal profiles.

The clear and co-varying redistribution of *bcd* and Stau observed here indicates transport of a *bcd* + Stau complex, adding to earlier work on *bcd* RNA gradient formation [38, 44]. We observe this co-localization continuing into the 13[th] and early 14[th] cc.

**2.2.2 Pre-blastoderm to syncytial blastoderm.** The strong changes seen in the Cleavage stage are not as evident as in the next stage, the syncytial blastoderm. The posterior spread from cc10 to cc12 is about half of that found within Cleavage (Fig 4C). 2-exponential

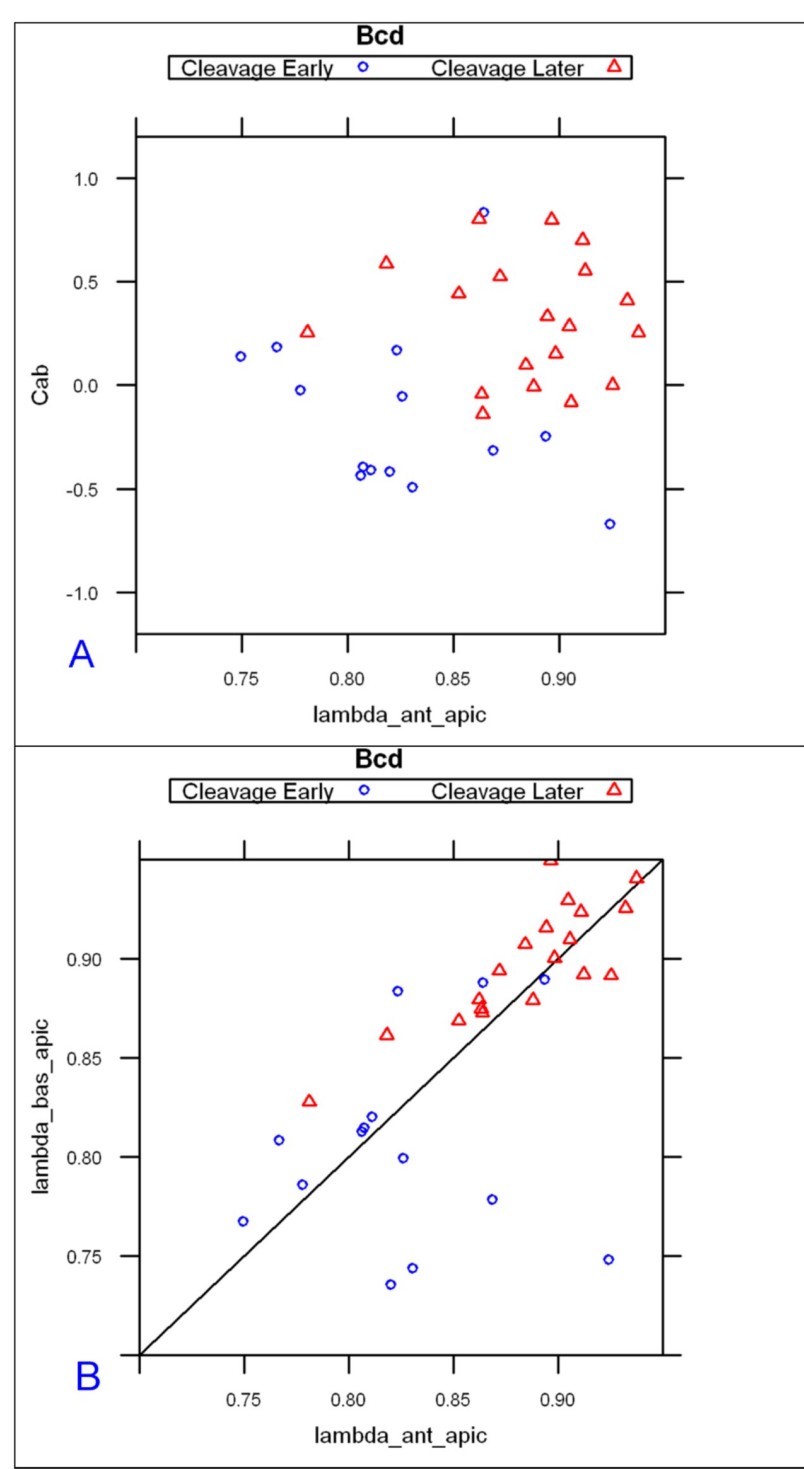

**Fig 7.** *bcd* **mRNA profiles from cleavage early are clearly distinguishable from Cleavage Late in scatterplots of** $\lambda_{\mathbf{anterior}}^{\mathbf{(apical)}}$ **vs** $\mathbf{C^{ab}} = \mathbf{In}\left(\mathbf{C_{anterior}^{(apical)}}/\mathbf{C_{anterior}^{(basal)}}\right)$ **(A), and** $\lambda_{\mathbf{anterior}}^{\mathbf{(apical)}}$ **vs** $\lambda_{\mathbf{anterior}}^{\mathbf{(basal)}}$ **(B).** The diagonal line in (B) is $\lambda_{\mathrm{anterior}}^{\mathrm{(apical)}} = \lambda_{anterior}^{(basal)}$. Data fit from (10,80) %EL.

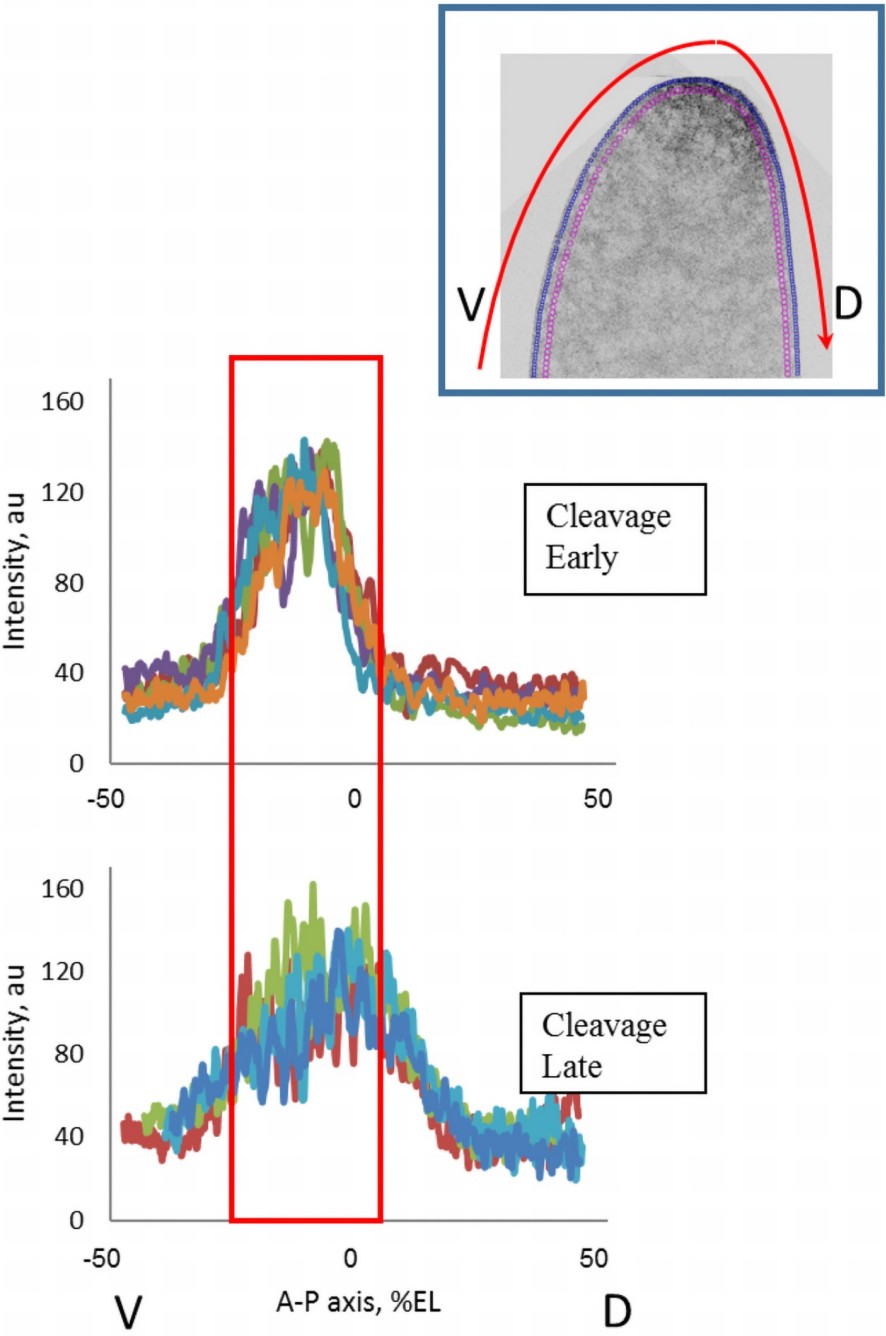

**Fig 8. Early cleavage (ClErl) embryos have very compact anterior *bcd* mRNA distributions; this becomes more extended posteriorly by later cleavage (ClLat).** Profiles extracted from ventral (V) through dorsal (D), through the anterior pole (see red arrow in inset).

parameters show more overlap for syncytial blastoderm than for Cleavage (Fig 10), with later stages generally showing more spread out gradients and lower apical intensity (see also Fig 4C for mean parameter values). S4 Fig compares Cleavage with Early Syncytial Blastoderm (cc10-12). The change between pre-blastoderm and syncytial blastoderm stages is primarily an increase in the $\lambda_{\text{anterior}}^{(\text{apical})}$ parameter.

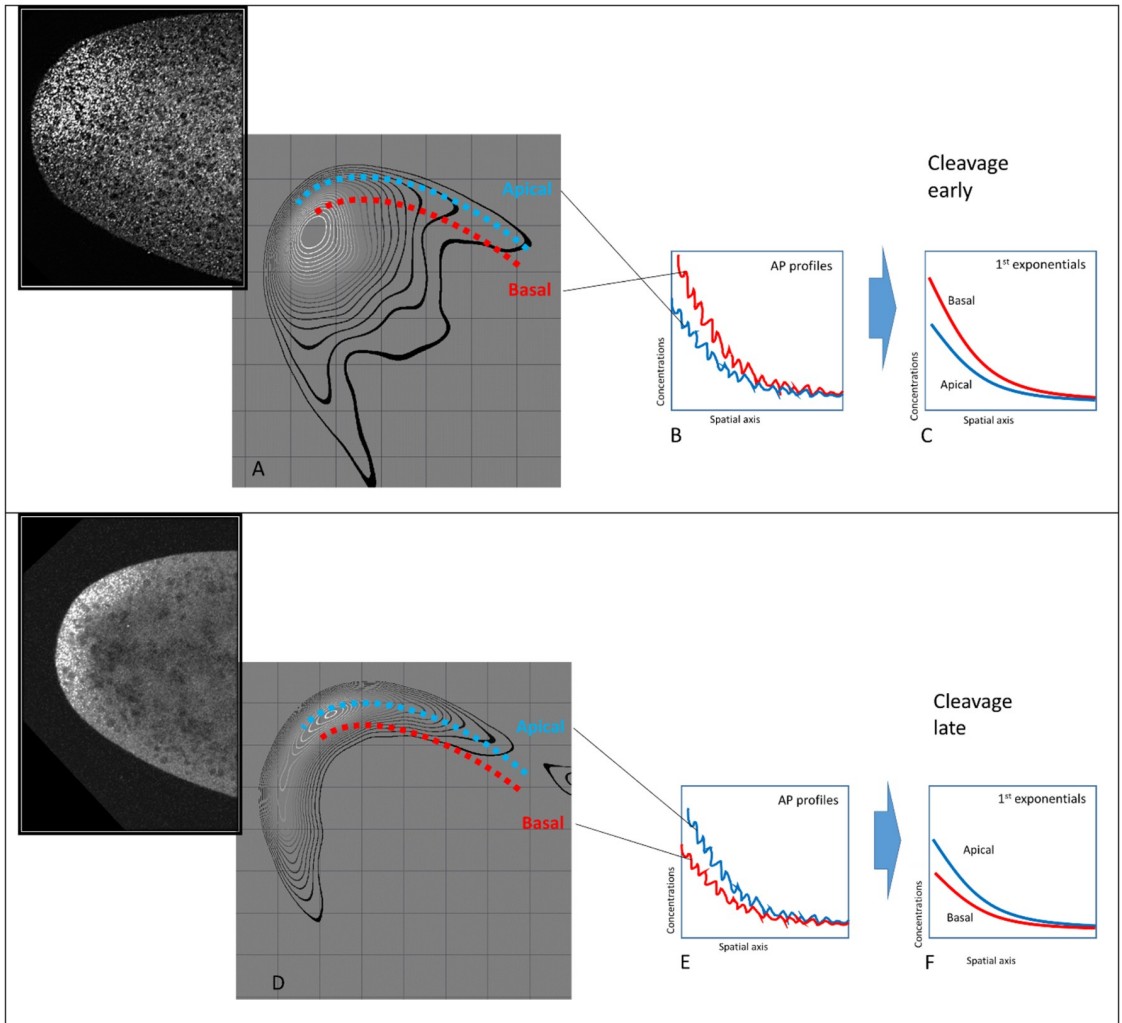

**Fig 9. Correspondence between 2D analysis and 1D exponential parameters in showing *bcd* (and Stau) redistributions in the early (pre-blastoderm stage) embryo.** When the bcd+Stau complexes are mainly in the interior of the head (with less material in the cortex, A), $C_{anterior}^{(apical)} < C_{anterior}^{(basal)}$ (B to C). When bcd+Stau spreads to the posterior (D), $\lambda_{anterior}^{(apical)}$ and $\lambda_{anterior}^{(basal)}$ increase (both exponentials became shallower, E to F). The insets in A and B show raw *bcd* images of in the head of the embryo.

**2.2.3 Syncytial blastoderm to cellularization.** We again observe strong changes in the two-exponential parameters going into cellularization (cc14). In this case, the changes in developmental time are in the apical:basal intensity ratio; the lambda is relatively fixed at this time (i.e. the posterior-wards spread is halted). The developmental trajectory shows the apical intensity increasing steadily over this period (Fig 4C, for mean values). Fig 11 shows a clear separation of cc10-13 and cc14 groups by apical intensity, for both bcd mRNA and Stau (indicating coordinated underlying biology). This is a reversal of the decreasing apical intensity seen from Cleavage to syncytial blastoderm (Fig 4).

*2.2.3.1 Signal redistribution in cc13.* The 26 embryos in the cc13 dataset are a large enough sample size to compare changes in 1D parameter profiles with redistributions visible in the 2D mid-sagittal plane. During cc 10–12, both bcd mRNA and Stau appear in a bright thin strip of the head cortical cytoplasm (not shown). Within even 12 minutes into cc13, it appears that these distributions are changing: in some images, both bcd mRNA and Stau are compact in

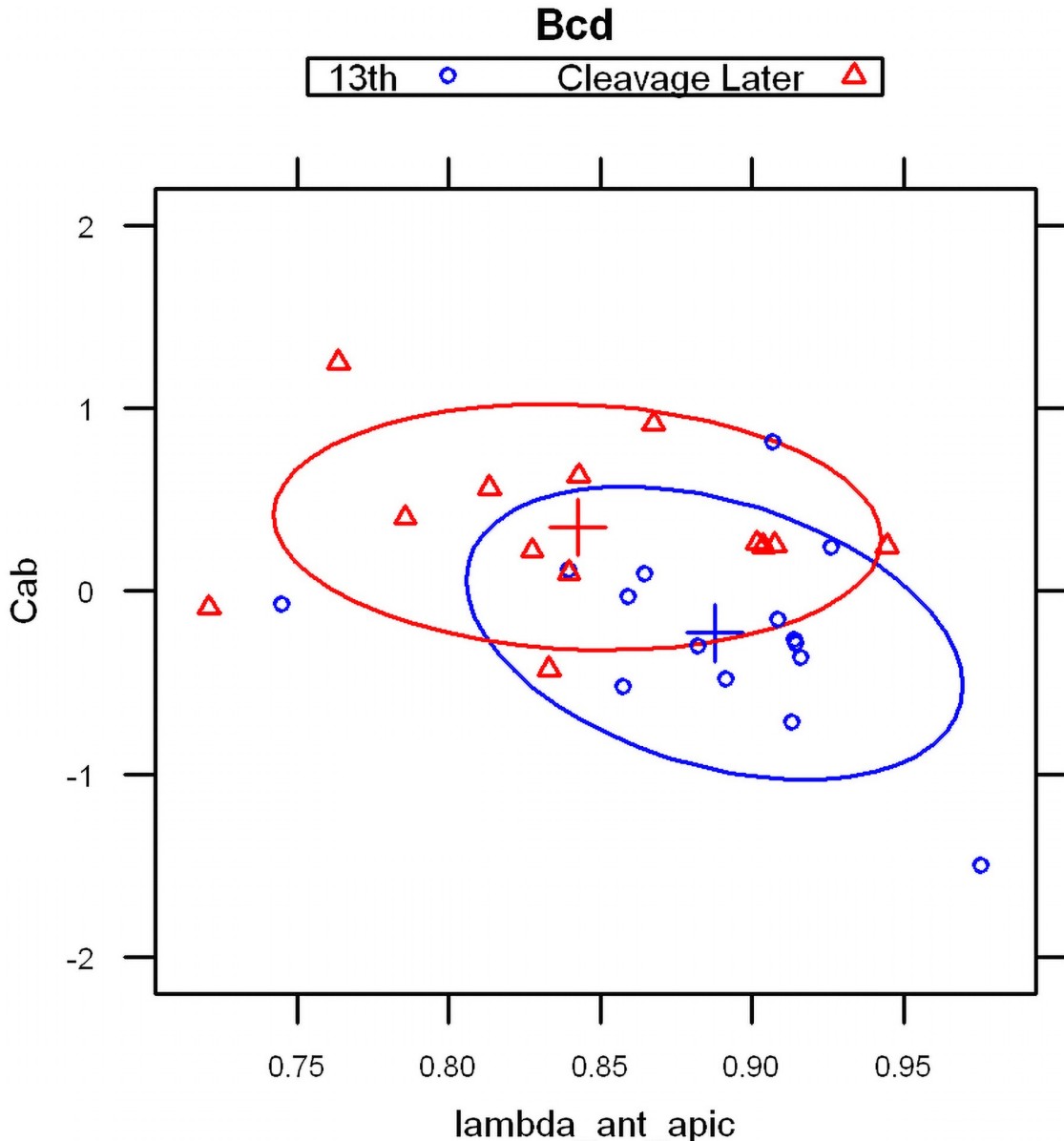

**Fig 10. 2-exponential parameters ($\lambda_{anterior}^{(apical)}$ vs $C^{ab} = \text{In}\left( C_{anterior}^{(apical)}/C_{anterior}^{(basal)} \right)$) for embryos from cleavage late and syncytial blastoderm cc13 stages.** Overlap is more common in syncytial blastoderm stages (clustering towards the upper left of the scatterplot).

the anterior but positioned deeply, below the nuclear layer ($C_{anterior}^{(apical)} < C_{anterior}^{(basal)}$); other images show more apical distributions, in the sub-nuclear and nuclear layers, and are more broadly spread out (Fig 12; cf Figs 9 and 14). The data does not allow us to temporally order these patterns.

*2.2.3.2 Redistribution in early cc14.* In our previous study [36], we observed large redistributions of bcd mRNA at the beginning of cc14, with fast basal to apical transport, followed by fast apical degradation (cf [24]). Analysis of this phenomenon is complicated by the short time window involving, 15–20 minutes only. Now, with a larger bcd mRNA sample size in this

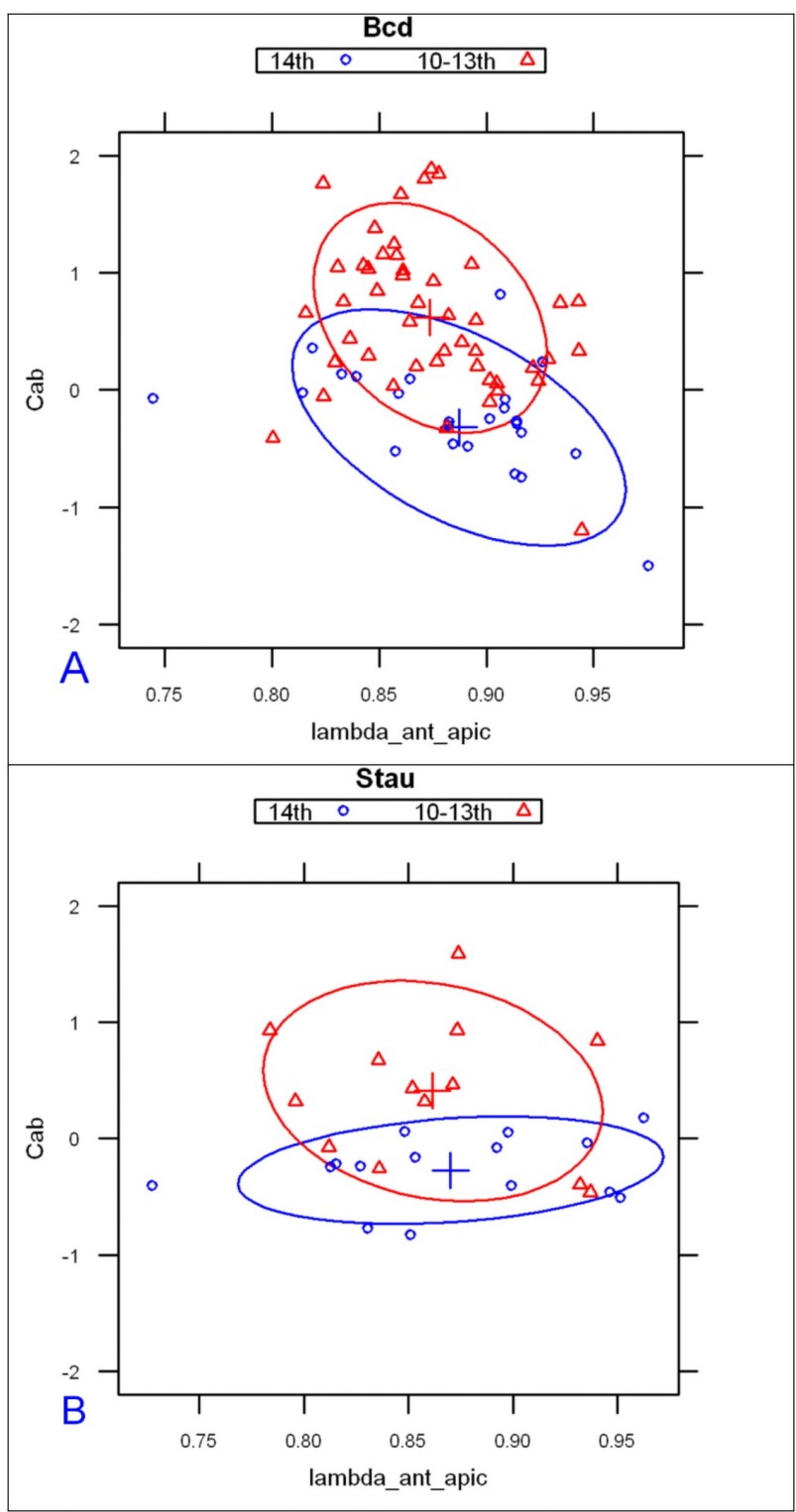

**Fig 11. Apical intensities (C1 parameters) rise sharply from syncytial blastoderm (red) to cc14 (blue; cellularizing blastoderm).** This is seen with both *bcd* mRNA (A) and Stau (B). This is a 'flip' from the decreasing apical intensity seen earlier, from the cleavage to syncytial blastoderm stages.

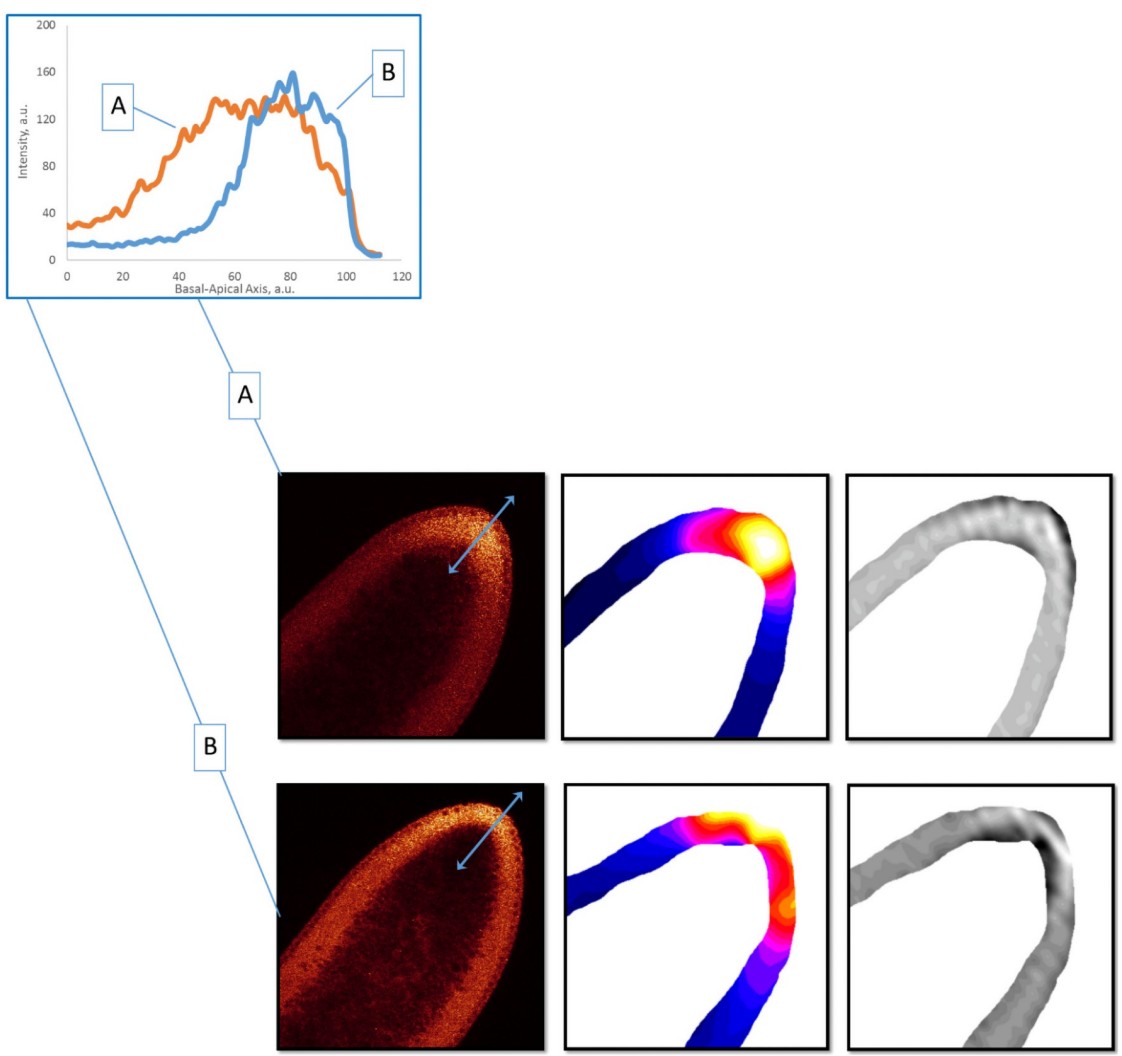

**Fig 12. Stau distributions in cortical cytoplasm in cc 13.** Raw images (1st column). 2D SSA decomposition highlights these differences in signal (2nd column); the 3rd column shows the residual noise in the SSA decomposition. Inset: corresponding 1D profile changes in the basal-apical direction (two-headed arrows on A and B).

period, we are able to subdivide profiles from the first few minutes of cc14 into earliest (Early1) and subsequent (Early2) stages by careful visual inspection, using DICOM software [65] which allowed us to simultaneously map intensities in colors, and to check developmental stage according to nuclear morphology [36]. Visual comparison of Early1 and Early2 embryos shows apical profiles becoming brighter than basal along the whole embryo length, while averaged basal profiles remain unchanged. This is also reflected in the apical:basal ratio from the 2-exponential fit (Fig 13): the difference between the Early1 ratio of 1.487 and the Early2 ratio of 2.678 is statistically significant. This provides a means for automatically tracking baso-apical redistribution (likely reflecting fast basal to apical transport [36]). See also S5 Fig for further comparison of cc13 and early cc14 sub-stages. These results indicate that the transition to high apical intensity occurs in the first minutes of cc14.

For Stau, while sample sizes are insufficient for statistics such as Fig 13, 2D-patterns show similar features. In particular, we observe compact anterior localization of Stau at all basal-

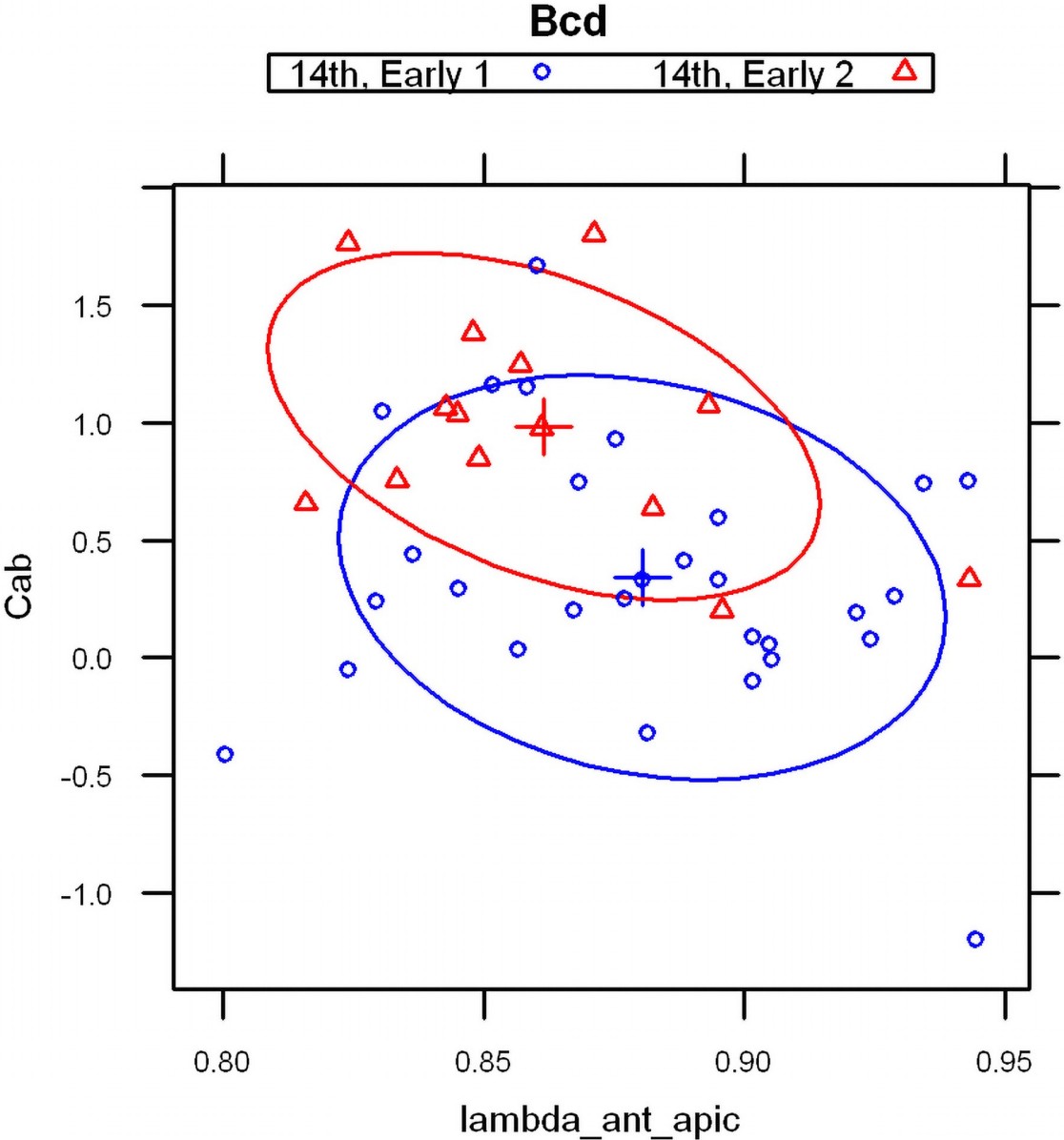

**Fig 13. Redistribution of *bcd* mRNA at the beginning of cc14, scatterplot of $\lambda_{\text{anterior}}^{(\text{apical})}$ vs $C^{\text{ab}} = \text{In}\left(C_{\text{anterior}}^{(\text{apical})}/C_{\text{anterior}}^{(\text{basal})}\right)$.** Within several minutes, the apical intensity rises above the basal.

apical depths in Early1 (Fig 14A and 14B, column II). Early2 embryos show posterior and apical movement from this earliest pattern (Fig 14C and 14D), with Stau forming a thin apical layer (short basal-apical gradient) at the stage when *bcd* mRNA is also undergoing basal to apical transport. The concerted basal to apical redistribution of *bcd* mRNA (Fig 13) and Stau (Fig 14) in the first minutes of cc14 likely reflects a joint redistribution of *bcd* + Stau complexes.

In summary, the trends of the 1D exponential parameters and movements in the 2D sagittal plane indicate a joint redistribution of *bcd* mRNA and Stau from head apical cortex layers into the interior of the head cytoplasm (and back) several times during early development.

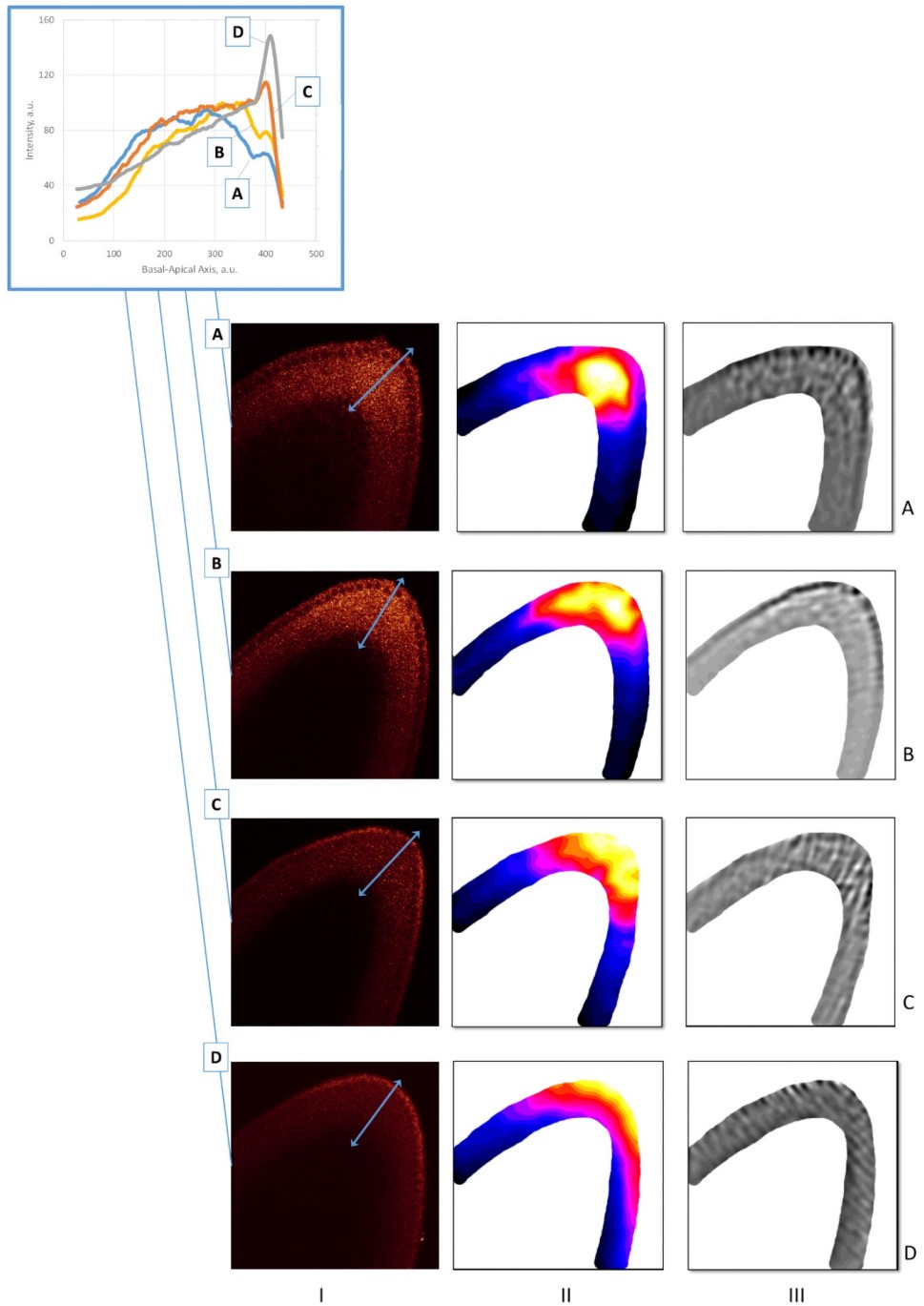

**Fig 14. The redistribution of Stau in early cc14.** Embryos A & B are characteristic of the Early1 stage; embryos C & D are characteristic of Early2. *shaped* 2D SSA [66] was used to process the signal from the cortical cytoplasm. The 1st column are the raw Stau scans, the 2nd column are the visualization of the 1st component of the 2D SSA decomposition, F1, and the 3rd column are the visualization of the mid-frequency components (F2 –F4; reflecting inhomogeneities from local cytoplasmic flows, local compartmentalization events, the processes of cellularization, etc.). The inset shows the profiles of the A-D embryos (the directions of the profiles are shown by light-blue two-headed arrows on the A-D images, column I), with a general trend of up and out for Stau movement in early cc14.

## 3 Discussion

Early fly embryo patterning is mainly governed by the primary morphogenetic gradient of the Bcd protein. But the mother supplies the zygote not with this protein factor, but with the *bcd* mRNA. The mRNA, begins to be translated in the late oocyte [38], but full-scale translation unfolds after fertilization. Two major events follow fertilization: the mRNA + Stau containing particles release and synchronized nuclear divisions begin. Our 2-exponential and 2D SSA analysis appears to have captured some of the large redistributions of mRNA+Stau particles related with the spatial distribution of the Bcd-production areas in these post-fertilization events. The most informative parameters from the 2-exponential modelling are in the anterior component, comparing $C_{anterior}^{(apical)}$ vs $C_{anterior}^{(basal)}$ and $\lambda_{anterior}^{(apical)}$ vs $\lambda_{anterior}^{(basal)}$.

First, the anterior exponential intensities ($C_{anterior}^{(apical)}$, $C_{anterior}^{(basal)}$; Fig 7A) start out quite different, but the apical exponent becomes stronger over developmental time, corresponding to the basal-apical redistribution of *bcd* mRNA in the head region. We also find that Stau follows the redistributions seen in *bcd*, and observe this in both the 1D exponential intensities and changes in the 2D patterning (Figs 12 and 14).

Second, we note a gradual increase of $\lambda_{anterior}^{(apical)}$ from the cleavage stage to cc13 (Figs 4C and 10), corresponding to the continued extension of the Bcd production area.

Third, we detect distinct baso-apical redistribution of *bcd* mRNA and Stau at the very beginning of cc14, in terms of changes in $C_{anterior}^{(apical)}$, $C_{anterior}^{(basal)}$ (Figs 4C and 11), quantifying earlier observations by [36, 39].

Biologically, these observations for *bcd* RNA and Stau reflect key stages of primary gradient formation (i), progressive intensification (ii), and disassembly (iii).

The variability in exponential parameters is likely to reflect both intrinsic noise as well as highly dynamic changes at each stage of early development (cf Figs 11, 12, 14 and 16). 2D-SSA analysis indicates distinct contributions of pre-cellular cytoplasmic organization to the early patterning and noise, which can stem from both biological variability and experimental errors (Cf [61, 67, 68]).

We find that the spatial extent of the *bcd* mRNA source is comparable with the Bcd protein gradient, as noted previously [36]. However, conclusions about the relation between *bcd* RNA and Bcd protein profiles depend strongly on the estimation of the non-specific fluorescence background in confocal microscopy. These different estimates can support different conclusions regarding the mechanism of Bcd gradient formation. We address this effect of estimation in more detail below.

### 3.1 The morphogen gradient source formation

Just after fertilization *bcd*+Stau particles are released and are soon observed in core plasm, forming a cone or wedge distribution (see [44] S1A, S1B Fig & S8 Fig & [38] Figs 6 and 7). Much of the *bcd* mRNA is not near the cortex but is found in the embryo interior [44]. Following the 6[th] mitosis, the RNP material is again observed at the anterior cortex and spreads along the cortex, forming a cup-shaped distribution ([44] S9E–S9H Fig, [38] Figs 6, 9, Cf our S3 Fig). Little et al. [44] observed that nuclei penetrate the cloud of *bcd* mRNA during their cortical migration, by the start of the syncytial blastoderm stage. The nuclei moving into the anterior *bcd*-rich area probably become involved in the highly effective and adjustable Bcd-production area.

These redistributions of *bcd* approximately coincide in time with the processes of active migration of nuclei in the core cytoplasm towards the head. At some point, these nuclei and *bcd*-containing granules become spatially overlapping [44].

It would be expected for the anterior Stau signal to be similar to *bcd* (since it is involved in *bcd* RNP complex transport), and this is what we observe (Figs 9 and 12 and S2 Fig). Similarities include material redistribution from the mainly apical cortical localization to the inner part of the head and backward movement to its periphery beneath the cortex (Fig 9A and 9B and S2 Fig), as seen in both intensity profiles and the apical and basal anterior exponential parameters (Fig 9C and 9F). The 1D parameter changes correspond to changes seen in the 2D SSA expression surfaces (Cf Fig 7).

Little et al. [44] suggested the presence of as yet uncharacterized structures along which *bcd* mRNA particles might translocate upon egg activation, arguing that since particles do not move further to the posterior after cc 3 that *bcd* mRNA remains tethered to underlying cytoskeletal structure(s) throughout embryogenesis (cf [21]). This correlates with earlier discussions of cellular structures associated with *bcd* mRNA redistribution, such as P bodies: Weil and co-authors proposed that translational activation of *bcd* mRNA resulted from disassembly of P bodies upon egg activation [26]. Cai and co-authors [39] also mention an invisible barrier to the *bcd* + Stau movement in the core cytoplasm of the unfertilized egg.

Our quantitative analysis corroborates earlier observations on the complicated redistributions of *bcd* in pre-syncitial blastoderm embryos [7, 36, 38–40, 44].

In the present work, we have also observed these processes with the Stau data and characterized these quantitatively.

## 3.2 Periodic rearrangements of the Bcd production area

Our data (both with 1D exponential analysis and 2D surfaces from SSA) allowed us to observe at least four periods in which *bcd* and Stau are redistributed: immediately after fertilization (1); during the early cleavage stage (2); during cc 13 (3); and during early cc 14 (4). Distributions characteristically move along the B-A axis from the surface cortical layer inwards, to compact interior head localization, and then back to a more posteriorly spread cortical distribution. (cf Fig 9 with Figs 12 and 14). These processes have been described for *bcd* [39, 44]. The observations by Cai with co-authors [39] that Stau co-localizes with *bcd* in these redistributions, are confirmed here by quantitative analysis. The co-localization strongly indicates the involvement of Stau with *bcd* in active transport.

What the mechanisms are that actively move material from surface layers deep into the cytoplasm and back is an open question. We can note that these movements appear to coincide in time and directions with cytoplasmic fountain streaming and retraction (cf [69]). These streams could potentially be redistributing *bcd* + Stau particles.

## 3.3 How extended is the bcd source?

The above discussion focused on comparing the $C_0$ and $\lambda$ parameters of the 1st anterior, head exponent at different developmental stages. The 2nd shallow posterior exponent is less informative for the developmental stage, but can be indicative of the extent of the *bcd* mRNA source compared to the Bcd protein gradient.

The quantitative analysis of the large *bcd* mRNA and Stau dataset in this paper corroborates earlier observations that the extent of the mRNA morphogen source could be comparable with that of the Bcd protein gradient (Fig 15; see also [36]).

It is well recognized that quantitative features of the Bcd morphogenetic gradient can depend crucially on how background fluorescence is treated (and is relevant to both mRNA and protein measurements). The simplest hypothesis of a flat background was used in both Little et al. [44], and Spirov et al. [36] (there, using *bcd* at 95%EL as the background value, as illustrated in Fig 15A and 15B). As Grimm with co-authors [28] noted, methods of computing the

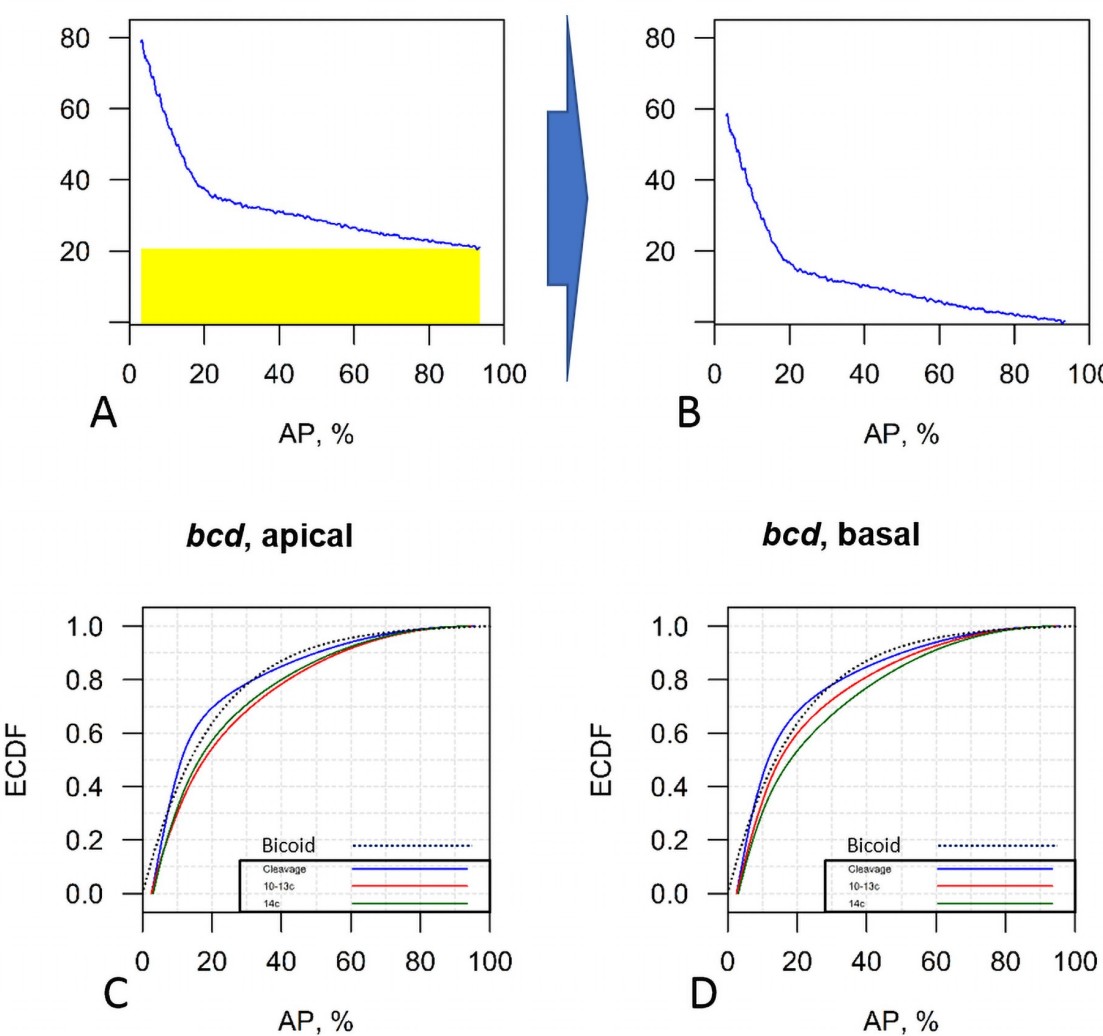

**Fig 15. Cumulative distribution of the *bcd* mRNA fluorescence intensity critically depends on the background definition (cumulative plot of Fig 5 data).** (A-B) The simplest hypothesis of a flat background (A) produces a *bcd* mRNA profile which is comparable to the Bcd protein gradient up to the beginning of cycle 14 (B). (C-D) Profiles from Fig 5 had uniform background subtracted, and are then plotted as cumulative curves. The blue dotted line is the approximation of a Bcd gradient with the length constant = 20%EL (see text). ECDF—empirical cumulative distribution function.

length constant strongly depend on estimates of the background level of fluorescence. For this situation, the *bcd* RNA cumulative curves (background subtracted) are quite close to a Bcd morphogen gradient with the common approximation of the *length constant* $\equiv -1/\alpha = 20\%$ (reviewed in [28]), (Fig 15C and 15D, blue dotted line). If the flat background is subtracted, we find length constants for the RNA profiles (from 0.15 to 0.25) are also comparable to the Bcd protein value of ~ 20%EL). These similar protein and RNA decays could suggest shared and/or coupled transport for *bcd* and Bcd (or a reduced need for fast Bcd diffusivity, if it is being more locally translated from an extended mRNA source).

Little et al. [44] proposed that *bcd* RNP complexes form an anterior gradient which is too short to be exclusively responsible for the geometry of the Bcd morphogen gradient. It is possible, however, that the RNA containing particles become smaller as more *bcd* RNA is involved in the translation machinery to produe Bcd protein. In other words, as *bcd* RNA becomes active, it goes beneath the detection limit for the Little et al. approach. Another possibility is that posterior bcd mRNA has a longer poly A tail. This would provide stronger *bcd* translation per *bcd* RNP particle.

Further experiments to quantify the background signal would help to resolve the real biological shape of the *bcd* mRNA gradient. As is, the form of the background produces uncertainty in extracting biological information such as the co-extent of the two gradients.

### 3.4 The Bcd gradient is still puzzling

In the present work, we have introduced an approach for quantifying *bcd* mRNA and Stau distributions in the early fly embryo (from post-fertilization release at the apical cortex to early cc14 when fast b*cd* degradation begins). This provides a method for tracking developmental changes in distribution, which corroborates earlier observations in a quantitative way. The results are summarized in Fig 16.

In this article, based on the results of the analysis of our data, we formulate a hypothesis about periodic redistributions of bcd + Stau complexes as one of the mechanisms of active transport of *bcd* mRNA. We treat this hypothesis as the development and detailing of the ARTS model. It is based on our quantitative analysis of the profiles and surfaces of *bcd* expression in comparison with those for Stau, and links our observations with the conclusions of the recent publication [40]. As demonstrated by Cai with coauthors [40], the mechanisms of *bcd* transport are much more complex than is presumed.

For the first time, such a redistribution of bcd + Stau is clearly observed during the period from fertilization through 9 cc (Fig 16a). This complex is redistributed from the cortical layers of the head to the core plasma and back to the cortical layers. As a consequence, the complex eventually spreads out to the posterior, giving a shallower spatial gradient than initially. We observed similar redistributions in the 13th and the beginning of the 14th cc (Fig 16c and 16d). This results in the further posterior spreading of the spatial gradient of the bcd + Stau complex. With our dataset, we cannot distinguish whether similar movements occur in cycles 10–12 (Fig 16b). Comparing our conclusions with the results of other authors, we can suggest that bcd + Stau are capable of binding to certain cell structures (primarily Golgi and ER), apparently through MT (Fig 16 inset a-I). It is noteworthy that the elements of the Golgi complex occupy the anterior cytoplasm of the early embryo head [70] (as well as ER [71]), and it is there that MT are observed, as well as MT-binding protein CLASP / Chb in complex with these MT [40].

We can also compare bcd + Stau redistributions to observations of periodic cytoplasmic streaming in the early embryo head [69]. Namely, that first cytoplasmic streaming in the form of a reverse fountain moves the material from the periphery of the head inward, deep into the core plasm (Fig 16 inset a-II). Cytoplasmic streaming then moves the material from the core back to the peripheral layers of the head by a (direct) fountain. These correlated movements suggest that cytoplasmic streaming participates in the redistribution and increasing extent of bcd + Stau. We suspect later bcd + Stau movements may also be correlated with cytoplasmic movements, but this requires further analysis.

There remain many unanswered questions regarding early events in *bcd* + Stau redistribution, but it is likely that these movements are involved with mRNA encountering nuclei and the initiation of translation. Further unanswered questions that our approach to quantifying

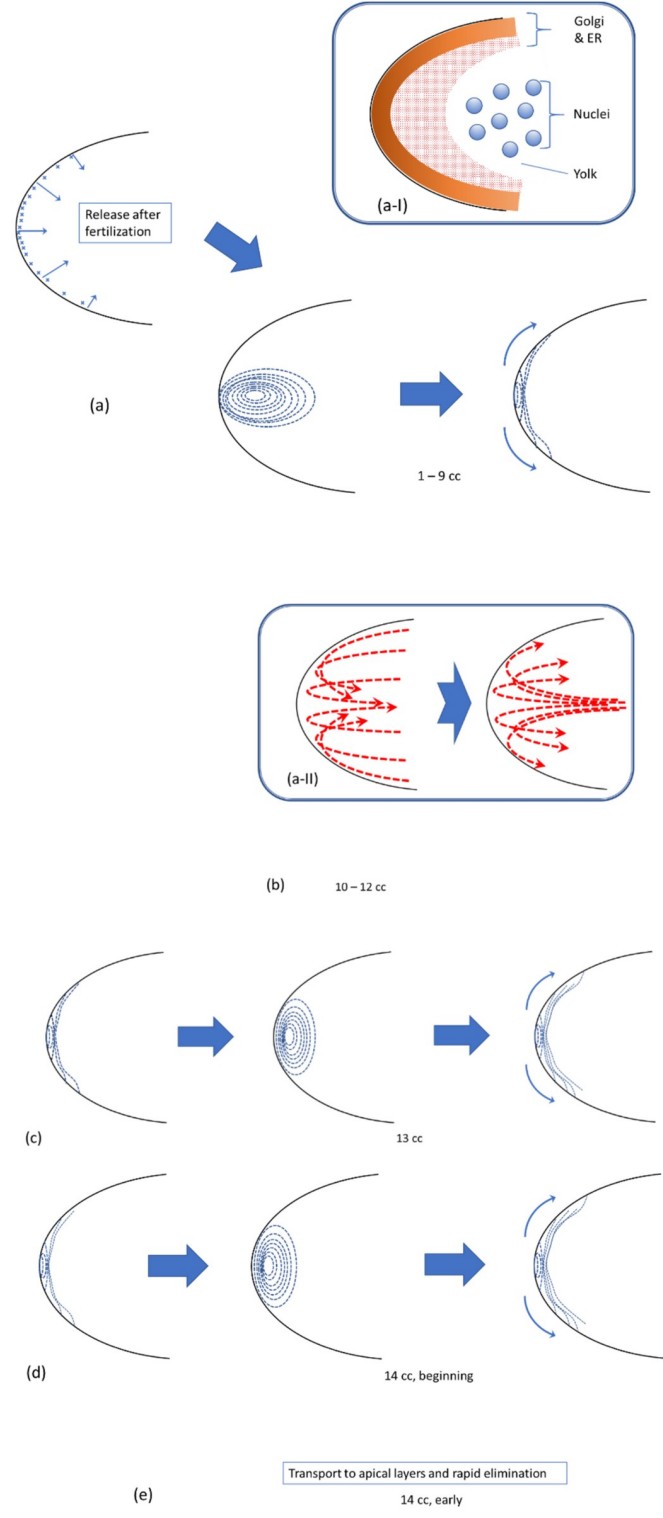

**Fig 16. The dynamics of bcd mRNA (in complex with Stau) for the early fly embryo, from its release at the apical cortex (a) as a result of fertilization to the stage of early cycle 14 (e), when fast bcd degradation occurs.** See text for details.

data can help include: Why does the *bcd* + Stau complex form a cone inside the core plasm of the early embryo, thinning towards the posterior direction (see [36, 44] S1A and S1B Fig & S8 Fig and [38] Figs 6 and 7)? Why do these cone intrusions appear in pairs (S3 Fig)? Why does inactivation of the cortical system through null-mutation of αTub67C tubulin or strong *ncd* mutation lead to greater activity of the core transport [37]? Why, under a number of mutations (e.g. kavarnull/Df(3L)55 and ncd1/ncd1) which increase *bcd* flow into the core, does the mRNA form a cloud-shaped distribution which spreads further towards the posterior (Cf. [37] Fig 4C and 4F; [39])? And finally, what is the role of microfilaments in these events?

## 4 Conclusions

- Qualitative and quantitative analysis of 1D profiles and expression surfaces obtained from confocal microscopy of early fly embryos is complicated by multiple experimental and biological sources of variability and noise. Data processing with singular spectrum analysis can be an effective tool for processing this data and clearly separating it into signal and noise components.

- The mathematical analysis of one-dimensional data (expression profiles) corresponds well to observations on two-dimensional distributions (expression surfaces).

- We observe highly dynamic *bcd* and Stau distributions in the early fly embryo. Our approach quantifies previous qualitative observations of *bcd* and Stau co-localization. This supports the hypothesis of active transport of the mRNA.

- Our quantitative approach reveals several *bcd* + Stau redistribution episodes, starting from cortical localization, moving into the deeper layers of the head cytoplasm, and then moving back to the apical cortex.

- Differences in data processing and insufficient tissue preservation leading to loss of apical cytoplasm in [44] may be the source for discrepancies between recent publications regarding the steepness of the *bcd* mRNA gradient ([36] vs [44]).

## 5 Methods

### 5.1 FISH, staining by immunohistochemistry and data acquisition

Fluorescence in situ hybridization (FISH) for *bcd* mRNA, immunohistochemistry for Stau and confocal imaging are as described in [36]. Intensity profiles were obtained from a circular area (ROI) moved along the dorsal cortex of midsagittal sections (Fig 17A), as described in detail in a separate methodological paper [60].

### 5.2 Dataset & considered stages of embryo development

The dataset (from the Baumgartner lab) consists of 160 embryos stained for *bcd* mRNA and 55 embryos stained for Stau, from unfertilized eggs (not considered) to later cycle 14A. In addition, about two dozen embryos stained for Stau were scanned at higher magnification (in the head half only) and used for 2D SSA analysis. The *bcd* mRNA dataset was previously used in [36].

### 5.3 Two-exponential modeling

The details of the two-exponential modelling on the example of the *bcd* mRNA A-P profiles, including the approach's accuracy and efficacy, is described in a separate methodological

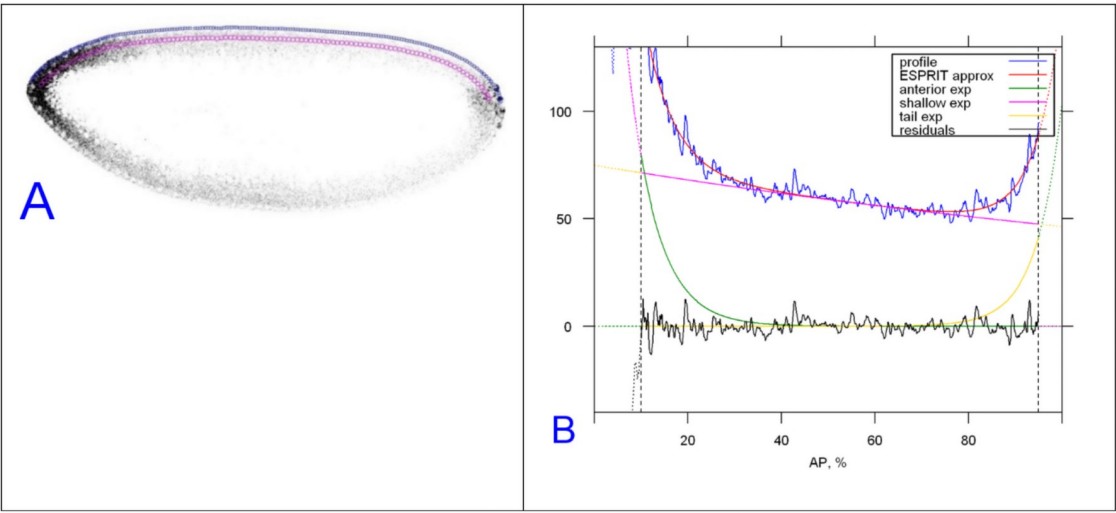

**Fig 17. Acquisition of intensity profiles for anteroposterior direction.** An example of extracting the anteroposterior profile of Staufen protein (same process used for *bcd* mRNA) from mid-cycle 14, and its decomposition as three exponentials. (A) sagittal image of Stau immunostaining (greyscale) with the apical (blue) and basal (magenta) ROI chains. (B) decomposition of the apical [prof 2] Stau profile into three exponential functions: strong anterior (green), moderate posterior (yellow) and nearly linear background (magenta).

paper [60]. This includes the choice of model-based characteristics which are independent of the microscopy settings and details of using the method to study biological profiles.

### 5.4 2D SSA and shaped 2D SSA

2D SSA is our extension of SSA approach to image processing [72, 73]. Details and further extensions of the approach (e.g. shaped 2D-SSA) was described in a methodological paper [66]. In the current work, 2D SSA and shaped 2D-SSA were applied directly to the raw pixel data.

## Supporting information

**S1 File.**
(DOCX)

**S1 Fig. The high variability of real experimental data on example of 18 apical and basal bcd profiles for early cleavage embryos (1–6 cc).**
(TIF)

**S2 Fig. The redistribution of the bcd mRNA (A, B) and Stau (C, D) during preblastoderm stage.**
(TIF)

**S3 Fig. The wedge-like paired intrusions of the Stau inside the head cytoplasm, visualized by 2D SSA technique.**
(TIF)

**S4 Fig. The averaged two-exponential models for two cleavage sub-groups (cleavage early & cleavage late), in comparison with the Early Syncytial Blastoderm embryos (the 10–12 cc) for bcd mRNA.**
(TIF)

**S5 Fig. The key differences between 13th cycle and early 14th cycle both for the bcd mRNA profiles and for the Stau factor profiles.**
(TIF)

## Acknowledgments

We thank the reviewers whose comments and suggestions helped improve and clarify this manuscript.

## Author Contributions

**Conceptualization:** Nina Golyandina, Alexander V. Spirov.

**Data curation:** Stefan Baumgartner.

**Formal analysis:** Alex Shlemov, Theodore Alexandrov, Nina Golyandina.

**Funding acquisition:** Stefan Baumgartner, Alexander V. Spirov.

**Methodology:** Nina Golyandina.

**Project administration:** Alexander V. Spirov.

**Software:** Alex Shlemov, Theodore Alexandrov.

**Supervision:** Alexander V. Spirov.

**Visualization:** Alex Shlemov, Theodore Alexandrov, Nina Golyandina.

**Writing – original draft:** Nina Golyandina, David Holloway, Alexander V. Spirov.

**Writing – review & editing:** David Holloway, Stefan Baumgartner.

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
