## [Decision Letter · Decision Letter 0]

3 Feb 2021

PONE-D-20-39109

Quantification reveals early dynamics in Drosophila maternal gradients

PLOS ONE

Dear Dr. Spirov,

Thank you for submitting your manuscript to PLOS ONE. After careful consideration, we feel that it has merit but does not fully meet PLOS ONE’s publication criteria as it currently stands. Therefore, we invite you to submit a revised version of the manuscript that comprehensively addresses the points raised during the review process.

We look forward to receiving your revised manuscript.

Kind regards,

Michael Schubert

Academic Editor

PLOS ONE

Reviewers' comments:

Reviewer's Responses to Questions

**Comments to the Author**

1. Is the manuscript technically sound, and do the data support the conclusions?

Reviewer #1: Yes

Reviewer #2: Yes

Reviewer #3: Yes

2. Has the statistical analysis been performed appropriately and rigorously? 

Reviewer #1: N/A

Reviewer #2: I Don't Know

Reviewer #3: I Don't Know

3. Have the authors made all data underlying the findings in their manuscript fully available?

Reviewer #1: Yes

Reviewer #2: Yes

Reviewer #3: Yes

4. Is the manuscript presented in an intelligible fashion and written in standard English?

Reviewer #1: Yes

Reviewer #2: Yes

Reviewer #3: Yes

5. Review Comments to the Author

Reviewer #1: I have read this paper with interest. The authors make a long introduction aiming to summarize what is known from the establishment of the gradients of bicoid protein and mRNA. The main result of this paper is the positive correlation between the positional localization of bicoid mRNA and Stau protein.

However, I have some scientific concerns about some assertions made in the paper:

1. On page 3, the sentence “Cells (or their nuclei) can sense local concentrations of morphogen molecules and respond by activating particular genes at given concentration thresholds (Wolpert, 1969; Crick, 1970). This results in a differential morphogen concentration being translated into a pattern of gene expression.” There is no proof that cells, genes or any other mechanism are sensitive to thresholds. This is a popular image associated to the so called French flag model and can be interpreted as a metaphor. This has no explicative function neither adds noting to the text. So I propose: “.... embryonic pattern formation. Cells (or their nuclei) can sense local concentrations of morphogen molecules and respond by activating particular genes at given concentration thresholds (Wolpert, 1969; Crick, 1970). This results in a differential morphogen concentration being translated into a pattern of gene expression. Since the theory was first developed, a number of morphogenetic gradients... “ change to “.... embryonic pattern formation. A number of morphogenetic gradients ... fly (Drosophila) embryo (References to the work of Nusslein-Volhardt and co-workers)”. By the way, the concept of thresholds could be understood as a bifurcation phenomenon (R. Dilão and D. Muraro, A software tool to model genetic regulatory networks. Applications to the modeling of threshold phenomena and of spatial patterning in Drosophila, PLoS ONE, 5 (5) (2010) 1-10 (e10743).

2. The SSD model is a toy model that does not explain nothing about the mechanisms about the bicoid gradient development. Also the argument of the so called 3 exponentials model used in this paper is strange, not to say ridiculous. I can invent, as a joke, a number of functions that fit the data. I can give fancy names to these functions. I don’t understand why experienced researchers use these argumentation tricks to try to justify observed data without the reference to the chemical and physical processes involved. The alternative to the SSD model --- the mRNA diffusion model --- was first proposed by R. Dilão, D. Muraro, M. Nicolau and M. Schoenauer, Validation of a morphogenesis model of Drosophila early development by a multi-objective evolutionary optimization algorithm. In C. Pizzuti, M.D. Ritchie, and M. Giacobini (eds.), “Evolutionary Computation, Machine learning and Data Mining in Bioinformatics”, Lecture Notes in Computer Science Vol. 5483, pp. 176–190, 2009. And followed by: R. Dilão and D. Muraro, mRNA diffusion explains protein gradients in Drosophila early development, Journal of Theoretical Biology, 264 (2010) 847-853, DOI:10.1016/j.jtbi.2010.03.012; and R. Dilão, Bicoid mRNA diffusion as a mechanism of morphogenesis in Drosophila early development, Comptes Rendus - Biologies, 337 (2014) 679-682, DOI: 10.1016/j.crvi.2014.09.004. The first paper is prior to the discovery of mRNA diffusion in Drosophila by Spirov and co-authors. In all these papers, it is discussed the similarities and dissimilarities between the protein and mRNA diffusion. There is also a paper relevant for this discussion which is F. Alves and R. Dilão, Modeling segmental patterning in Drosophila: maternal and gap genes, Journal of Theoretical Biology, 241 (2006) 342-359.

3. In page 7, the authors write “While a number of modelling projects address details and processes not yet quantified experimentally [Dilao - Muraro, 2010 47; Kavousanakis, et al., 2010 48; Deng et al., 2010; Little et al., 2011; Liu et al., 2011 49; Dalessi et al., 2012 50; Shvartsman, Baker, 2012 51; Liu, Niranjan, 2011 52; Liu, Niranjan, 2012 53]), many are indicating that the extended mRNA source is important, and that the SDD model is too simple.” SSD is based on nothing, in Dilao - Muraro, 2010, the model with mRNA is calibrated and a mechanism is given. The authors in the current paper are also authors on the paper of Cai & al, saying that the SSD is void. I think that “too simple” is an euphemism for “wrong”. On the other hand, the authors of this paper use a 3 exponential model based on SSD.

4. And there are more papers where the mRNA diffusion model has been explored and calibrated with experimental data of FlyEx.

5. This is a strange way of doing science.

Reviewer #2: Formation and interpretation of morphogen gradients are two important aspects of embryonic patterning. For decades, formation of the Bicoid (Bcd) gradient and its role in patterning the anterior-posterior axis of Drosophila have served as model systems to study both aspects. Early models of Bcd gradient formation postulated a simple model of translation and free diffusion of Bcd proteins from a localized source of bcd mRNAs deposited at the anterior pole of the embryo. More recent data, however, indicated that an active transport mechanism actually mediates the formation of a bcd mRNA gradient with the help of a dedicated machinery that involves the protein Stau, a process deemed important in forming a properly functioning Bcd protein gradient. The exact mechanism for how this happens is current unknown.

In this manuscript, Shlemov et al uses a signal analysis framework previously developed by (some of) the authors to analyse the dynamics of bcd and Stau distributions in space and time in the early Drosophila embryo, hoping to gain insights into Bcd gradient formation mechanisms. While the authors report some novel observation on the distribution dynamics of these factors, they fail to engage with their data to shed more light on the process, rendering the manuscript intellectually thin, and unfit for publication in a research journal like plos one.

Reviewer #3: This insightful and important contribution identifies components of the complex multi-stage process that distributes Bcd/Staufen in the early Drosophila embryo. I am not competent to judge the mathematical analysis and therefore limit the following comments to biological aspects. I congratulate the authors for their success in clarifying this very challenging (and contentious) topic. Suggestions are minor.

The exacting mathematical analysis relies on the degree to which the Bcd and Staufen distributions are preserved in the preparations and to the histological/imaging methods that were used for quantification. It might help readers if the authors specify the cell cycles they refer to by the descriptors pre-blastoderm, syncytial blastoderm and cellularization (or instead of using these terms?), and specify the actual numbers of embryos at each nuclear cycle that were analyzed. Also, because embryos are not radially symmetric and quantification is better for optical planes closer to the microscope objective, whether data showed sensitivity to the orientation of embryos on the microscope slide.

The authors point out the importance of background subtraction methodology and might consider whether background in different locations (cortex vs interior) might not be comparable because, for example, components such as yolk and lipid granules are not uniformly distributed and their distinct contributions to background might differ.

P25 “Following the 6th mitosis, the RNP material moves back to the anterior cortex and spreads along the cortex, forming a cup-shaped distribution” This statement is based on observed steady state distributions without evidence that the RNP material actually moves. Perhaps a more accurate statement would simply describe the different distributions at the different stages?

P25 “At some point, these nuclei and bcd containing granules collide [Little at al., 2011].” Is “collide” accurate? Is “spatial overlap” a better way to describe the fact that cytoplasm that surrounds the nuclei expands to encompass cytoskeletal elements that are not restricted to nuclear islands?

The authors state that Bcd protein production initiates at fertilization but it is my understanding that the Ali-Murthy study showed that Bcd protein is present in stage 14 oocytes and therefore that Bcd in the early embryo represents a combination of protein produced either before and after fertilization.

6. PLOS authors have the option to publish the peer review history of their article (what does this mean?). If published, this will include your full peer review and any attached files.

Reviewer #1: **Yes: **Rui Dilão

Reviewer #2: No

Reviewer #3: No

---

## [Author Response · Author response to Decision Letter 0]

19 Apr 2021

Dear Editors,

Thank you very much for the review of our manuscript entitled: “Quantification reveals early dynamics in Drosophila maternal gradients”. 

We sincerely appreciate all of the valuable comments and suggestions, which helped us to improve the quality of the article. Our responses to the Reviewers’ comment are described below point-to-point. Changes suggested by the Reviewers have been made to the manuscript (highlighted within the document). 

Reviewer #1: 

I have read this paper with interest. The authors make a long introduction aiming to summarize what is known from the establishment of the gradients of bicoid protein and mRNA. The main result of this paper is the positive correlation between the positional localization of bicoid mRNA and Stau protein.

However, I have some scientific concerns about some assertions made in the paper:

1. On page 3, the sentence “Cells (or their nuclei) can sense local concentrations of morphogen molecules and respond by activating particular genes at given concentration thresholds (Wolpert, 1969; Crick, 1970). This results in a differential morphogen concentration being translated into a pattern of gene expression.” There is no proof that cells, genes or any other mechanism are sensitive to thresholds. This is a popular image associated to the so called French flag model and can be interpreted as a metaphor. This has no explicative function neither adds noting to the text. So I propose: “.... embryonic pattern formation. Cells (or their nuclei) can sense local concentrations of morphogen molecules and respond by activating particular genes at given concentration thresholds (Wolpert, 1969; Crick, 1970). This results in a differential morphogen concentration being translated into a pattern of gene expression. Since the theory was first developed, a number of morphogenetic gradients... “ change to “.... embryonic pattern formation. A number of morphogenetic gradients ... fly (Drosophila) embryo (References to the work of Nusslein-Volhardt and co-workers)”. 

Agreed. Done (on page 3). Those references have been added also. 

By the way, the concept of thresholds could be understood as a bifurcation phenomenon (R. Dilão and D. Muraro, A software tool to model genetic regulatory networks. Applications to the modeling of threshold phenomena and of spatial patterning in Drosophila, PLoS ONE, 5 (5) (2010) 1-10 (e10743). 

We are grateful to have pointed to this article, which we have now cited. 

2. The SSD model is a toy model that does not explain nothing about the mechanisms about the bicoid gradient development. Also the argument of the so called 3 exponentials model used in this paper is strange, not to say ridiculous. I can invent, as a joke, a number of functions that fit the data. I can give fancy names to these functions. I don’t understand why experienced researchers use these argumentation tricks to try to justify observed data without the reference to the chemical and physical processes involved. 

Our approach is focused on the phenomenological quantitative description of expression profiles for their quantitative analysis and classification. This approach is based on the application of the mathematical subspace-based method ESPRIT as described in the methodological article [Alexandrov et al., 2018]. Interpretation of the simple math model in physico-chemical terms is the ultimate goal outlined in the Discussion (please see our revised section “3.4 The Bcd gradient is still puzzling.” And the revised Figure 16). Specifically, the 3-exponential model is proposed as a possible extension of the 2-exponential model. This is a special case when the posterior exponential component is clearly observable. This posterior component, as can be assumed, corresponds to the Stau complexes with maternal mRNAs localized posteriorly (oskar, first of all). The 3rd component is discussed with respect to posterior pattern; but the analysis in this article is based on the 2-exponential fit. As we consider in the Discussion, this paper, together with the publication [Cai et al., 2019], demonstrate that the processes of bcd mRNA redistribution, as seen with the component dynamics, are much more complex than is commonly thought. Biologically, these include coupled processes of (active) transport, immobilization in the Golgi / ER complex (for translation), and periodic events caused by synchronous mitoses. The ESPRIT decomposition of the data is a means of quantifying expression that can capture dynamic changes during development. 

The alternative to the SSD model --- the mRNA diffusion model --- was first proposed by R. Dilão, D. Muraro, M. Nicolau and M. Schoenauer, Validation of a morphogenesis model of Drosophila early development by a multi-objective evolutionary optimization algorithm. In C. Pizzuti, M.D. Ritchie, and M. Giacobini (eds.), “Evolutionary Computation, Machine learning and Data Mining in Bioinformatics”, Lecture Notes in Computer Science Vol. 5483, pp. 176–190, 2009. And followed by: R. Dilão and D. Muraro, mRNA diffusion explains protein gradients in Drosophila early development, Journal of Theoretical Biology, 264 (2010) 847-853, DOI:10.1016/j.jtbi.2010.03.012; and R. Dilão, Bicoid mRNA diffusion as a mechanism of morphogenesis in Drosophila early development, Comptes Rendus - Biologies, 337 (2014) 679-682, DOI: 10.1016/j.crvi.2014.09.004. The first paper is prior to the discovery of mRNA diffusion in Drosophila by Spirov and co-authors. In all these papers, it is discussed the similarities and dissimilarities between the protein and mRNA diffusion. There is also a paper relevant for this discussion which is F. Alves and R. Dilão, Modeling segmental patterning in Drosophila: maternal and gap genes, Journal of Theoretical Biology, 241 (2006) 342-359. 

We thank the reviewer for pointing these appropriate citations. We have included the key ones (Dilão et al., 2009; Dilão, Muraro, 2010a; Dilão, Muraro, 2010b; Dilão 2014) at the appropriate places in our article. 

3. In page 7, the authors write “While a number of modelling projects address details and processes not yet quantified experimentally [Dilao - Muraro, 2010 47; Kavousanakis, et al., 2010 48; Deng et al., 2010; Little et al., 2011; Liu et al., 2011 49; Dalessi et al., 2012 50; Shvartsman, Baker, 2012 51; Liu, Niranjan, 2011 52; Liu, Niranjan, 2012 53]), many are indicating that the extended mRNA source is important, and that the SDD model is too simple.” SSD is based on nothing, in Dilao - Muraro, 2010, the model with mRNA is calibrated and a mechanism is given. The authors in the current paper are also authors on the paper of Cai & al, saying that the SSD is void. I think that “too simple” is an euphemism for “wrong”. 

Some authors of this MS have repeatedly discussed the failure of the SSD model (e.g., Baumgartner, S. 2018. Seeing is believing: the Bicoid protein reveals its path. Hereditas, 155), and such criticism is not the main purpose of this article. 

On the other hand, the authors of this paper use a 3 exponential model based on SSD. 

We disagreed. The 2 (!!) exponential model is phenomenological one and not based on SDD, as noted above. The methodology here is based on the use of ESPRIT for the phenomenological description of the profiles. 

4. And there are more papers where the mRNA diffusion model has been explored and calibrated with experimental data of FlyEx. 

We used our original approaches and our original dataset to thoroughly and reproducibly characterize quantitatively the behavior of the bcd mRNA (as described in [Spirov et al., 2009]). The dataset is complemented by a set of Stau confocal scans. The data were specially collected by new experimental approaches in order to achieve their best quality and best reproducibility. 

5. This is a strange way of doing science. 

Reviewer #2: 

Formation and interpretation of morphogen gradients are two important aspects of embryonic patterning. For decades, formation of the Bicoid (Bcd) gradient and its role in patterning the anterior-posterior axis of Drosophila have served as model systems to study both aspects. Early models of Bcd gradient formation postulated a simple model of translation and free diffusion of Bcd proteins from a localized source of bcd mRNAs deposited at the anterior pole of the embryo. More recent data, however, indicated that an active transport mechanism actually mediates the formation of a bcd mRNA gradient with the help of a dedicated machinery that involves the protein Stau, a process deemed important in forming a properly functioning Bcd protein gradient. The exact mechanism for how this happens is current unknown.

In this manuscript, Shlemov et al uses a signal analysis framework previously developed by (some of) the authors to analyse the dynamics of bcd and Stau distributions in space and time in the early Drosophila embryo, hoping to gain insights into Bcd gradient formation mechanisms. While the authors report some novel observation on the distribution dynamics of these factors, they fail to engage with their data to shed more light on the process, rendering the manuscript intellectually thin, and unfit for publication in a research journal like plos one. 

We formulated our conclusions based on the results of our analysis more clearly and compared them with the results of other publications. We explicitly formulated the relationship of our results with the development and detailing of the ARTS model, with respect to what we are observing with the data. This material (three new paragraphs that are highlighted) has been added to section “3.4 The Bcd gradient is still puzzling.” We also revised Figure 16 in that section (added two insets) to more clearly illustrate our conclusions. 

In brief:

• We formulate a hypothesis about periodic redistributions of bcd + Stau complexes as one of the mechanisms of active transport of bcd mRNA. It is based on our quantitative analysis of the profiles and surfaces of bcd expression in comparison with those for Stau, and links our observations with the conclusions of the recent publication [Cai et al., 2019]. 

• Comparing our conclusions with the results of other authors, we can suggest that bcd + Stau are capable of binding to certain cell structures (primarily Golgi and ER), apparently through MT (Cf [Cai et al., 2019]). 

• At the same time, we drew attention to the classic observations of periodic cytoplasmic streaming in the embryo head, such that the streaming can also participate in the redistributions bcd + Stau. 

Reviewer #3: 

This insightful and important contribution identifies components of the complex multi-stage process that distributes Bcd/Staufen in the early Drosophila embryo. I am not competent to judge the mathematical analysis and therefore limit the following comments to biological aspects. I congratulate the authors for their success in clarifying this very challenging (and contentious) topic. Suggestions are minor.

The exacting mathematical analysis relies on the degree to which the Bcd and Staufen distributions are preserved in the preparations and to the histological/imaging methods that were used for quantification. It might help readers if the authors specify the cell cycles they refer to by the descriptors pre-blastoderm, syncytial blastoderm and cellularization (or instead of using these terms?), 

We have specified the nuclear / cell cycles corresponding to these stages of early embryogenesis (both in Abstract and on 10th page, first paragraph). 

and specify the actual numbers of embryos at each nuclear cycle that were analyzed. 

We have specified the numbers of analyzed embryos for each developmental stage (on 10th page, first paragraph). {(In total we had 38 bcd vs 15 Stau cleavage stage embryos, 43 bcd versus 22 Stau syncytial stage embryos, and 56 bcd versus 18 Stau cellulization stage embryos).} 

Also, because embryos are not radially symmetric and quantification is better for optical planes closer to the microscope objective, whether data showed sensitivity to the orientation of embryos on the microscope slide. 

All embryos taken into analysis had a clear orientation, always showing the dorsal and ventral side, hence defining a sagittal section. In order to obtain the mid-sagittal-most section, 6 optical sections were recorded starting from the side closer to the objective, estimated 50 μm before the mid-sagittal-most section and ending shortly after the middle of the embryo. Later, the image with the largest dimension was chosen for analysis, as it most likely represented a true mid-sagittal section. The procedure was already used and described in Spirov et al. [2009]. In this respect, the reviewer is right: it does not make sense to choose an optical plane behind the middle, as this location is no longer favorable for good signal output. 

The authors point out the importance of background subtraction methodology and might consider whether background in different locations (cortex vs interior) might not be comparable because, for example, components such as yolk and lipid granules are not uniformly distributed and their distinct contributions to background might differ. 

We work with sagittal cross-sections, like many other authors in this area (e.g. [Spirov et al., 2009; Little et al., 2011; Grimm et al., 2012]). We measure intensities only along the cortical layers of the cytoplasm. However, we do not compare quantitatively the intensity between the core plasm and the cortical layers. We believe that the simplest flat background model is the most reasonable for such data. Whereas, for lateral scans, the core of the image includes a background, apparently due to the yolk. Therefore, the background in this case is convex [Myasnikova et al., 2005]. However, we did not use lateral images in this work. 

P25 “Following the 6th mitosis, the RNP material moves back to the anterior cortex and spreads along the cortex, forming a cup-shaped distribution” This statement is based on observed steady state distributions without evidence that the RNP material actually moves. Perhaps a more accurate statement would simply describe the different distributions at the different stages? 

We revised the sentence accordingly (P.26). “Following the 6th mitosis, the RNP material is again observed at the anterior cortex and spreads along the cortex, forming a cup-shaped distribution.” 

P25 “At some point, these nuclei and bcd containing granules collide [Little at al., 2011].” Is “collide” accurate? Is “spatial overlap” a better way to describe the fact that cytoplasm that surrounds the nuclei expands to encompass cytoskeletal elements that are not restricted to nuclear islands? 

We revised the sentence accordingly (P.26). “At some point, these nuclei and bcd-containing granules become spatially overlapping [Little at al., 2011].” 

The authors state that Bcd protein production initiates at fertilization but it is my understanding that the Ali-Murthy study showed that Bcd protein is present in stage 14 oocytes and therefore that Bcd in the early embryo represents a combination of protein produced either before and after fertilization. 

Here are the relevant sentences in our text: “Early fly embryo patterning is mainly governed by the primary morphogenetic gradient of the Bcd protein. But the mother supplies the zygote not with this protein factor, but with the bcd mRNA.” (Discussion, 1st paragraph). 

We revised the sentence accordingly. « Early fly embryo patterning is mainly governed by the primary morphogenetic gradient of the Bcd protein. But the mother supplies the zygote not with this protein factor, but with the bcd mRNA. The mRNA begins to be translated in the late oocyte [Ali-Murthy, Kornberg, 2016], but full-scale translation unfolds after fertilization.» 

Once again, we thank you for the time you put in reviewing our paper and look forward to meeting your expectations. Since your inputs have been precious, in the eventuality of a publication, we would like to acknowledge your contribution explicitly. 

The authors

---

## [Decision Letter · Decision Letter 1]

18 May 2021

PONE-D-20-39109R1

Quantification reveals early dynamics in Drosophila maternal gradients

PLOS ONE

Dear Dr. Spirov,

Thank you for submitting your manuscript to PLOS ONE. After careful consideration, we feel that it has merit but does not fully meet PLOS ONE’s publication criteria as it currently stands. Therefore, we invite you to submit a revised version of the manuscript that addresses the points raised during the review process.

We look forward to receiving your revised manuscript.

Kind regards,

Michael Schubert

Academic Editor

PLOS ONE

Journal Requirements:

Reviewers' comments:

Reviewer's Responses to Questions

**Comments to the Author**

1. If the authors have adequately addressed your comments raised in a previous round of review and you feel that this manuscript is now acceptable for publication, you may indicate that here to bypass the “Comments to the Author” section, enter your conflict of interest statement in the “Confidential to Editor” section, and submit your "Accept" recommendation.

Reviewer #1: All comments have been addressed

Reviewer #2: All comments have been addressed

Reviewer #3: All comments have been addressed

2. Is the manuscript technically sound, and do the data support the conclusions?

Reviewer #1: Partly

Reviewer #2: Yes

Reviewer #3: Yes

3. Has the statistical analysis been performed appropriately and rigorously? 

Reviewer #1: No

Reviewer #2: Yes

Reviewer #3: N/A

4. Have the authors made all data underlying the findings in their manuscript fully available?

Reviewer #1: Yes

Reviewer #2: Yes

Reviewer #3: Yes

5. Is the manuscript presented in an intelligible fashion and written in standard English?

Reviewer #1: Yes

Reviewer #2: Yes

Reviewer #3: Yes

6. Review Comments to the Author

Reviewer #1: Fortunately, the authors eliminated from the introduction the reference to the French flag model.

There are statements in the abstract that are known from the time of Crick (1960). For example: “supporting that the distribution and dynamics of bcd mRNA are key factors in the formation and maintenance of the Bcd protein morphogenetic gradient”. This is the so called “central dogma of molecular genetics”.

The statement “The observed co-localization during redistribution of bcd and Stau may indicate the involvement of active transport.”. If there is active transport, the equations describing the distribution are not exponential. By the way, the exponential function is only compatible with a diffusion mechanism (not even reaction-diffusion).

It is now clear that the so called two or three exponential equation is a phenomenological fit and the analysis presented by the authors is irrelevant to validate biological mechanisms. Fits are not models and are not used for validation. A mathematical model passes by 3 stages: benchmarking, calibration and validation. None of this has been done.

The exponential functions do not justify the profiles of Bcd, cf., for example, figure 2a, of Houchmandzadeh, Wieschaus, Leibler, Nature 415 (2002) 798-802. Moreover, there is a large variability found in the tails of BCD in different individuals. This analysis has not been done for bcd.

I have some difficulty in extracting useful biological information from this paper. Anyhow, I recognize the strong effort of some of the biological oriented authors. I think that there are some misinterpretations about the role of mathematical models in biology.

Reviewer #2: The authors adequately addressed my concerns, and the manuscript is now fit for publication in plos one.

Reviewer #3: This insightful and important contribution identifies components of the complex multi-stage process that distributes Bcd/Staufen in the early Drosophila embryo. I congratulate the authors for their success in clarifying this very challenging (and contentious) topic. I am satisfied with the revision and recommend publication.

7. PLOS authors have the option to publish the peer review history of their article (what does this mean?). If published, this will include your full peer review and any attached files.

Reviewer #1: No

Reviewer #2: No

Reviewer #3: No

---

## [Author Response · Author response to Decision Letter 1]

8 Jul 2021

Dear Dr. Schubert,

Thank you very much for your and the reviewers’ work on reading our revised submission. We are glad that Reviewers 2 and 3 are satisfied with the revisions and feel that the work is ready for publication. We respond to Reviewer 1’s specific comments in this letter. There don’t appear to be indications for further revision of the manuscript itself, but we can do so if there are specific indications. 

6. Review Comments to the Author

Reviewer #1: Fortunately, the authors eliminated from the introduction the reference to the French flag model.

Done. 

There are statements in the abstract that are known from the time of Crick (1960). For example: “supporting that the distribution and dynamics of bcd mRNA are key factors in the formation and maintenance of the Bcd protein morphogenetic gradient”. This is the so called “central dogma of molecular genetics”.

This section was not to revisit the central dogma, but rather to highlight that Bcd protein gradient formation has been frequently studied in isolation from bcd mRNA spatial patterning or the temporal dynamics of the bcd mRNA. 

The statement “The observed co-localization during redistribution of bcd and Stau may indicate the involvement of active transport.”. If there is active transport, the equations describing the distribution are not exponential. By the way, the exponential function is only compatible with a diffusion mechanism (not even reaction-diffusion).

As mentioned below, the exponential fit is phenomenological. 

It is now clear that the so called two or three exponential equation is a phenomenological fit and the analysis presented by the authors is irrelevant to validate biological mechanisms. Fits are not models and are not used for validation. A mathematical model passes by 3 stages: benchmarking, calibration and validation. None of this has been done.

Correct, this study quantifies the data. This suggests the traditional SDD mathematical model is inadequate, but there is no new mathematical modelling involved. New models should be able to account for the dynamic shifts we report. The mathematical aspects of the (phenomenological) model are discussed, among other things, and the results are justified in [Alexandrov T., Golyandina N., Holloway D., Shlemov A., and Spirov A. 2018. Two-exponential models of gene expression patterns for noisy experimental data. J.Comp.Biol. 25:1220-1230]. (This ref is from the paper, ref [60], not new.) 

The exponential functions do not justify the profiles of Bcd, cf., for example, figure 2a, of Houchmandzadeh, Wieschaus, Leibler, Nature 415 (2002) 798-802. Moreover, there is a large variability found in the tails of BCD in different individuals. This analysis has not been done for bcd.

Bcd protein profiles are not analyzed in this paper. As discussed above, exponentials are the basis functions for the phenomenological quantification of the bcd mRNA data. Variability is apparent and addressed across n = 160 bcd profiles, e.g. in Figs. 3, 7, 10, 11, 13. 

I have some difficulty in extracting useful biological information from this paper. Anyhow, I recognize the strong effort of some of the biological oriented authors. I think that there are some misinterpretations about the role of mathematical models in biology.

This is a quantification of the bcd mRNA and Stau protein gradient data. 

Please let us know if there are any further questions. 

With best regards, 

Alexander Spirov

---

## [Editor Report · Decision Letter 2]

19 Jul 2021

Quantification reveals early dynamics in Drosophila maternal gradients

PONE-D-20-39109R2

Dear Dr. Spirov,

We’re pleased to inform you that your manuscript has been judged scientifically suitable for publication and will be formally accepted for publication once it meets all outstanding technical requirements.

Kind regards,

Michael Schubert

Academic Editor

PLOS ONE

---

## [Editor Report · Acceptance letter]

10 Aug 2021

PONE-D-20-39109R2 

Quantification reveals early dynamics in *Drosophila* maternal gradients 

Dear Dr. Spirov:

I'm pleased to inform you that your manuscript has been deemed suitable for publication in PLOS ONE. Congratulations! Your manuscript is now with our production department. 

Kind regards, 

on behalf of

Dr. Michael Schubert 

Academic Editor

PLOS ONE